# Feedback inhibition by a descending GABAergic neuron regulates timing of escape behavior in *Drosophila* larvae

Jiayi Zhu[1,2], Jean-Christophe Boivin[1,2], Alastair Garner[1,2], Jing Ning[1], Yi Q Zhao[1], Tomoko Ohyama[1,3]*

[1]Department of Biology, McGill University, Montreal, Canada; [2]Integrated Program of Neuroscience, McGill University, Montreal, Canada; [3]Alan Edwards Center for Research on Pain, McGill University, Montreal, Canada

**\*For correspondence:**
tomoko.ohyama@mcgill.ca

**Competing interest:** The authors declare that no competing interests exist.

**Abstract** Escape behaviors help animals avoid harm from predators and other threats in the environment. Successful escape relies on integrating information from multiple stimulus modalities (of external or internal origin) to compute trajectories toward safe locations, choose between actions that satisfy competing motivations, and execute other strategies that ensure survival. To this end, escape behaviors must be adaptive. When a *Drosophila melanogaster* larva encounters a noxious stimulus, such as the focal pressure a parasitic wasp applies to the larval cuticle via its ovipositor, it initiates a characteristic escape response. The escape sequence consists of an initial abrupt bending, lateral rolling, and finally rapid crawling. Previous work has shown that the detection of noxious stimuli primarily relies on class IV multi-dendritic arborization neurons (Class IV neurons) located beneath the body wall, and more recent studies have identified several important components in the nociceptive neural circuitry involved in rolling. However, the neural mechanisms that underlie the rolling-escape sequence remain unclear. Here, we present both functional and anatomical evidence suggesting that bilateral descending neurons within the subesophageal zone of *D. melanogaster* larva play a crucial role in regulating the termination of rolling and subsequent transition to escape crawling. We demonstrate that these descending neurons (designated SeIN128) are inhibitory and receive inputs from a second-order interneuron upstream (Basin-2) and an ascending neuron downstream of Basin-2 (A00c). Together with optogenetic experiments showing that co-activation of SeIN128 neurons and Basin-2 influence the temporal dynamics of rolling, our findings collectively suggest that the ensemble of SeIN128, Basin-2, and A00c neurons forms a GABAergic feedback loop onto Basin-2, which inhibits rolling and thereby facilitates the shift to escape crawling.

## eLife assessment

The aim of this **important** study is to functionally characterize neuronal circuits underlying the escape behavior in *Drosophila* larvae. Upon detection of a noxious stimulus, larvae follow a series of stereotyped movements that include bending of their body, rolling and crawling away. This paper combines quantitative behavioral analyses, cell-type specific manipulations, optogenetics, calcium imaging, immunostaining, and connectomic analysis to provide **convincing** evidence of an inhibitory descending pathway that controls the switch from rolling to fast crawling behaviors of the larval escape response.

## Introduction

Virtually all organisms on earth face the threat of being maimed or killed by one or more predatory organisms. Not surprisingly, when organisms encounter threat-associated stimuli, they exhibit a wide variety of escape responses appropriate to their biological construction and the specific predators within their ecological niche (*Burrell, 2017*; *Campagner et al., 2023*; *Chin and Tracey, 2017*; *Im and Galko, 2012*; *Peirs and Seal, 2016*). Typically, these escape responses consist of a sequence of simple actions. The roundworm *C. elegans*, for example, in response to a touch to its head, exhibits rapid backward locomotion coupled with a suppression of head movements, followed by a deep ventral bend (omega turn) and a 180 degree reversal in the direction of locomotion. This sequence allows the roundworm to escape from nematophagal fungi that cohabitate with it in organic debris (*Chalfie and Sulston, 1981*; *Chalfie et al., 1985*).

When *Drosophila melanogaster* larvae encounter noxious stimuli, such as the stimulation that accompanies an attempt by a parasitic wasp to penetrate the larval cuticle with its ovipositor, they exhibit an escape response consisting of an initial abrupt bending, followed by lateral rolling, and finally, rapid crawling (*Hwang et al., 2007*; *Ohyama et al., 2015*; *Onodera et al., 2017*; *Tracey et al., 2003*). Previous work has shown that noxious stimuli are primarily detected by class IV dendritic arborization neurons (Class IV neurons) located beneath the body wall (*Tracey et al., 2003*). More recent studies have identified several important components in the downstream nociceptive neural circuitry, particularly those involved in rolling (*Burgos et al., 2018*; *Dason et al., 2020*; *Hu et al., 2017*; *Hu et al., 2020*; *Imambocus et al., 2022*; *Kaneko et al., 2017*; *Ohyama et al., 2015*; *Takagi et al., 2017*; *Yoshino et al., 2017*). To date, however, the neural mechanisms that underlie the rolling-escape sequence, notably, the transition from rolling to crawling, have remained unclear.

In this study, we provide both functional and anatomical evidence that, bilateral descending neurons in the subesophageal zone (SEZ) of *D. melanogaster* larva, which comprise part of a neural circuit underlying rolling, a characteristic nocifensive escape response, potentially regulate the termination of rolling and subsequent transition to escape crawling. We show that these descending neurons, which we designate as SeIN128, are identical to those denoted previously as SS04185 (*Ohyama et al., 2015*), are inhibitory neurons that receive inputs from Basin-2 (a second-order interneuron upstream) and A00c (an ascending neuron downstream of Basin-2), and provide GABAergic feedback onto Basin-2. Together with behavioral analyses of rolling during systematic optogenetic manipulation of SeIN128 and Basin-2 activity, our findings suggest that an ensemble of neurons—SeIN128, Basin-2, and A00c—forms an inhibitory feedback circuit that inhibits rolling, which in turn facilitates the shift to escape crawling.

## Results

### SS04185 facilitates rolling termination and shortens the latency of crawling behavior in the escape responses

In a previous study, we showed that activation of all Basin neurons (Basin-1, -2, -3, and -4) induced rolling followed by fast crawling (*Figure 1A–D*; *Ohyama et al., 2015*). Here, we first examined whether optogenetic activation of all four Basins expressing the red-shifted opsin CsChrimson (using Basin-1–4 Gal4, i.e., R72F11-Gal4) could elicit the same behavior. Upon activation of all Basins, we observed rolling mostly within the first 5 s, followed by crawling (*Figure 1B* (top panel), *Figure 1C, D*). The crawling speed during activation of all Basins following rolling was ~1.5 times that of the crawling speed at baseline (*Figure 1D*; *Ohyama et al., 2015*).

To identify the neurons responsible for escape behavior (rolling and/or fast crawling), we conducted a behavioral screening of ~250 split Gal4 lines that were labeled in the central nervous system (CNS) when co-activated with all Basins. With respect to rolling, we found that activation of the split-Gal4 line, SS04185 (i.e., $w^{1118}$; $R54B01\text{-}Gal4^{AD}$; $R46E07\text{-}Gal4^{DBD}$), significantly reduced the probability of rolling when compared to activating only the Basins (*Figure 1B, C, E*, *Videos 1 and 2*) without affecting the crawling speed during stimulation (*Figure 1D*).

The likelihood of rolling upon co-activation of SS04185 neurons and Basins might decrease because activation of SS04185 neurons trigger other actions, such as crawling, head casting, hunching, or stopping, and not because they solely inhibit rolling evoked by Basins. To investigate this possibility, we examined the effect of activating only SS04185 and found that this did not induce any extra actions

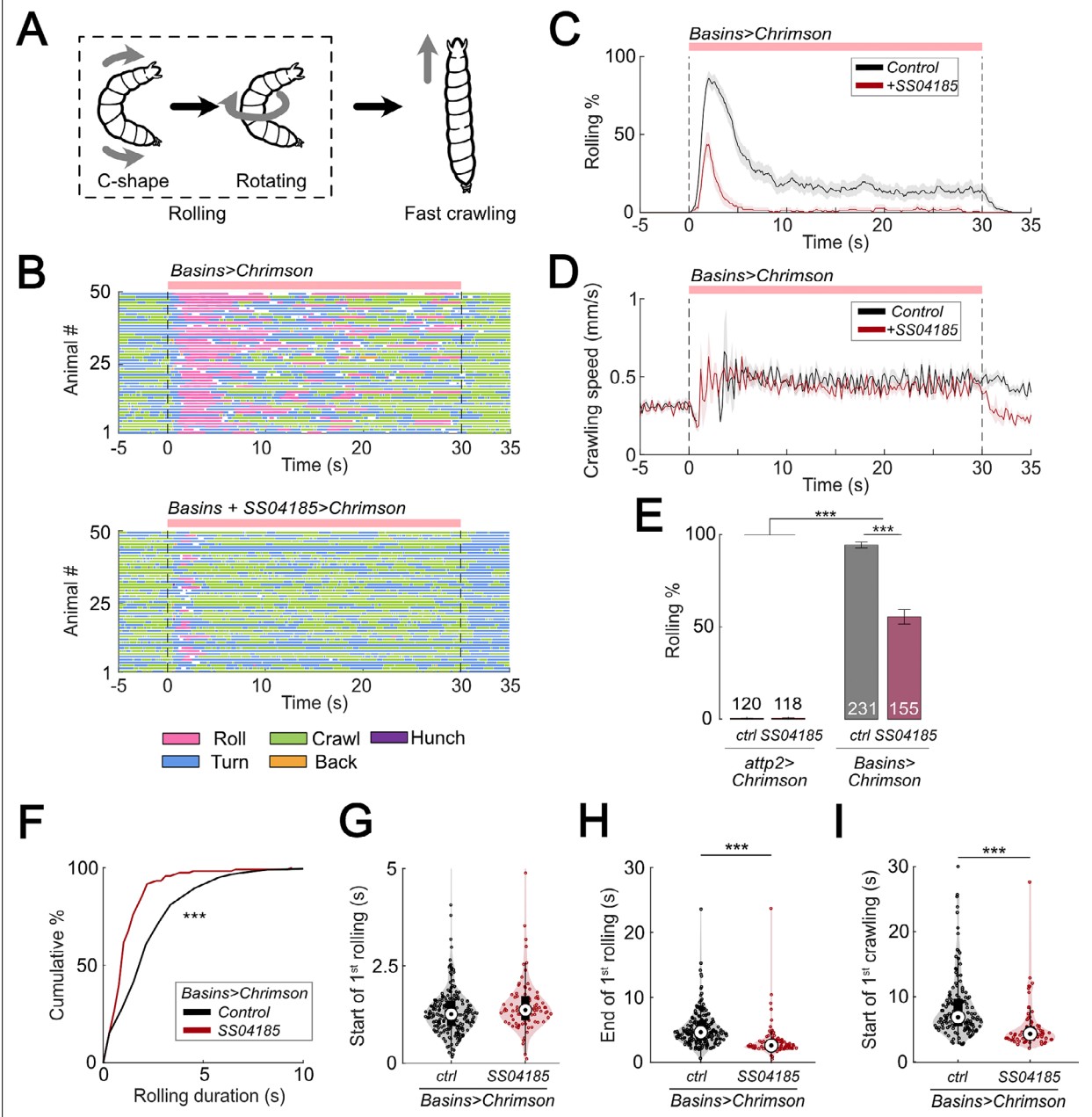

**Figure 1.** Activation of SS04185 inhibits rolling evoked by activation of Basin neurons. (**A**) Schematic of *Drosophila* larval escape behavior sequence. (**B**) Ethograms of Basin activation (top panel) and co-activation of SS04185 and Basins (bottom panel). Each row represents an individual larva. Pink, blue, green, orange, and purple lines represent bouts of rolling, turning, crawling, backward crawling, and hunching. The red bar and dashed lines indicate the time window during which neural activation was present. Genotypes: *20xUAS-IVS-CsChrimson::mVenus/+;+; R72F11-Gal4/+* (top); *20xUAS-IVS-CsChrimson::mVenus/+; R54B01-Gal4.AD/+; R46E07-Gal4.DBD/ R72F11-Gal4* (bottom). Genotypes in (**C, D, F–I**) are the same as those mentioned here. (**C**) Time series of larval crawling speed during co-activation of SS04185 and Basins (red) and activation of Basins alone (black). Shaded areas represent the standard error. The red bar and dashed lines denote the optogenetic stimulation window. (**D**) Time series of rolling probabilities of larvae during co-activation of SS04185 and Basins (red) and activation of Basins alone (black). Shaded areas represent 95% confidential intervals for rolling probabilities. The red bar and dashed lines denote the optogenetic stimulation window. (**E**) Rolling probabilities of larvae with activation of different neurons. Error bars represent the 95% confidence interval. Genotypes from left to right: (1) *20xUAS-IVS-CsChrimson::mVenus/+;;*, (2) *20xUAS-IVS-CsChrimson::mVenus/+; R54B01-Gal4.AD/+; R46E07-Gal4.DBD/+*, (3) *20xUAS-IVS-CsChrimson::mVenus/+;; R72F11-Gal4/+*, and (4) *20xUAS-IVS-CsChrimson::mVenus/+; R54B01-Gal4.AD/+; R46E07-Gal4.DBD/ R72F11-Gal4*. $n$ = 120, 118, 231, 155 from left to right. Statistics: Chi-square test, $\chi^2$ = 0, $p > 0.05$ for the first two groups; $\chi^2$ = 83.85, $p < 0.001$ for the last two groups; and $\chi^2$ = 365.51, $p < 0.001$ for the comparison between the first two groups and the last two groups. (**F**) Cumulative plot of rolling duration. Statistics: Mann–Whitney $U$ test, $p < 0.001$, $n$ = 652, 120. (**G**) A violin plot showing start of first rolling bout for each larva during stimulation. Statistics: Mann–Whitney $U$ test, $p$ = 0.027, $n$ = 225, 89. (**H**) A violin plot displaying end of first

*Figure 1 continued on next page*

*Figure 1 continued*

rolling bout for each larva during stimulation. Statistics: Mann–Whitney *U* test, p < 0.001, *n* = 225, 89. (I) A violin plot presenting start of first crawling bout for each larva during stimulation. Statistics: Mann–Whitney *U* test, p < 0.001, *n* = 214, 70. ***p < 0.001.

The online version of this article includes the following source data and figure supplement(s) for figure 1:

**Source data 1.** Excel sheet containing statistics data from behavior tracking depicted in *Figure 1E-I*.

**Source data 2.** Excel sheet containing raw data from behavior tracking depicted in *Figure 1E-I*.

**Figure supplement 1.** Activation of SS04185 inhibits rolling.

**Figure supplement 1—source data 1.** Excel sheet containing raw data from behavior tracking depicted in *Figure 1—figure supplement 1E, F and I*.

**Figure supplement 1—source data 2.** Excel sheet containing statistics data from behavior tracking depicted in *Figure 1—figure supplement 1E, F and I*.

**Figure supplement 2.** Activation of SS04185 inhibits rolling.

**Figure supplement 2—source data 1.** Excel sheet containing statistic data from behavior tracking depicted in *Figure 1—figure supplement 2C*.

**Figure supplement 2—source data 2.** Excel sheet containing raw data from behavior tracking depicted in *Figure 1—figure supplement 2B and C*.

such as crawling, turning, hunching, or stopping (*Figure 1—figure supplement 1A–D*). These data suggest that co-activation of Basins and SS04185 neurons reduces rolling because SS04185 activation inhibits the Basin circuit.

Next, we explored how the quality of rolling changed during co-activation of SS04185 and Basin neurons. First, we examined the amount of time animals spent rolling during Basin activation. The average time spent rolling (percentage of the 30 s stimulation period) was 23.9% (7.2 s out of 30 s) following activation of Basins alone, whereas it was only 5.9% following co-activation of Basins and SS04185 (1.8 s out of 30 s) (*Figure 1—figure supplement 1E*). Additionally, the duration of each rolling bout was significantly shorter when SS04185 neurons were co-activated with Basins (Mann–Whitney *U* test, p < 0.001; *Figure 1F*).

The duration of a rolling bout could decrease because of changes in the latency to initiate rolling, latency to terminate rolling, or both. To investigate how SS04185 activation affects these temporal parameters of rolling, we analyzed the latencies for the initiation and termination of the first rolling bout. Compared to activating Basins alone, co-activating the Basins and SS04185-expressing neurons only marginally increased latency to onset of the first rolling bout (*Figure 1G*), whereas it markedly

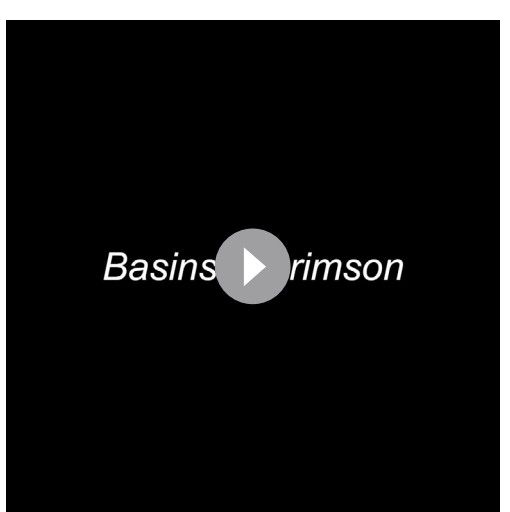

**Video 1.** *Drosophila* larva behavior during only Basins activation. Activation of Basins alone evokes protracted rolling followed by turning/crawling.
https://elifesciences.org/articles/93978/figures#video1

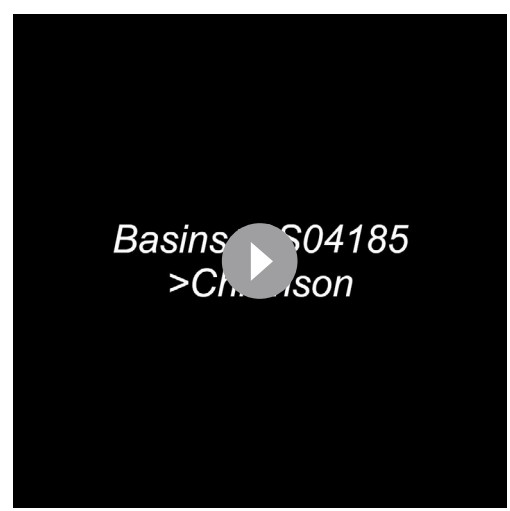

**Video 2.** *Drosophila* larva behavior during co-activation of Basins and SS04185 neurons. Co-activation of SS04185 and Basins evokes only brief rolling followed by turning/crawling.
https://elifesciences.org/articles/93978/figures#video2

reduced the latency for the termination of rolling (Mann–Whitney *U* test, p < 0.001; *Figure 1H*). These data strongly suggest that SS04185-expressing neurons are involved in terminating rolling.

If the rolling module inhibits crawling, then premature termination of rolling might allow crawling to commence sooner than normal. Co-activation of SS04185 and Basins resulted in the initiation of the first crawling bout occurring earlier than when only Basins were activated (Mann–Whitney *U* test, p < 0.001; *Figure 1I*, *Figure 1—figure supplement 1F*). The time from the end of rolling to the start of crawling remained similar between the groups in which the Basins were activated alone and in which the Basins and SS04185 were co-activated (*Figure 1—figure supplement 1G*). This is consistent with the higher probability of crawling during activation of SS04185 and Basin neurons (*Figure 1—figure supplement 1H*). Lastly, activation of SS04185 neurons in conjunction with Basins did not change the crawling speed compared to activation of Basins alone (*Figure 1—figure supplement 1I*). These results collectively indicate that SS04185 activation terminates rolling and facilitates the shift to fast crawling.

## A pair of descending neurons in SS04185 contributes to termination of rolling

To identify the neurons that express SS04185 upon CsChrimson activation, we examined the localization of SS04185-labeled neurons. We found that SS04185 split-Gal4 strongly labeled a pair of descending neurons located within the SEZ and mushroom body (MB) neurons within the brain (*Figure 2A*). To pinpoint which of these neurons are involved in reducing the probability of rolling (*Figure 1B, C, E*), we varied the level of SS04185 expression among the pair of SS04185-expressing descending neurons (SS04185-DN) and the SS04185-expressing MB (SS04185-MB) neurons (jointly with the Basins as in *Figure 1*). These manipulations allowed us to assess the resultant behavioral outcomes.

If SS04185-MB neurons are involved in the modulation of rolling, then reducing SS04185-MB expression should reduce the extent to which activation of both SS04185-DN neurons and SS04185-MB neurons decreases the probability of rolling. To test this conjecture, we expressed Killer Zipper (KZip$^+$), which interferes with the binding of Gal4$^{AD}$ and Gal4$^{DBD}$ in SS04185-MB neurons with MB LexA line (R13F02-LexA), consequently leading to a significant reduction in CsChrimson expression in SS04185-MB neurons (*Figure 2B*, *Figure 2—figure supplement 1A*; *Dolan et al., 2017*; *Vogt et al., 2016*). When compared to KZip$^+$ controls, which do not express SS04185 (*Figure 2C*, black bars), however, activation of SS04185 neurons with reduced SS04185-MB expression (*Figure 2C*, red bars on the right; *Figure 2—figure supplement 1B*) still reduced rolling probability (as well as the total duration of rolling [*Figure 2—figure supplement 1C*]) to a level no different from that of KZip$^-$ controls expressing SS04185 fully in both SS04185-MB and SS04185-DN neurons (*Figure 2C*, dark red bars in the middle). Additionally, co-activation of MB Gal4 lines (MB247-Gal4) with Basins (without activation of SS04185-DN neurons) did not reduce the probability of rolling (*Figure 2—figure supplement 1D, E*). These data indicate that SS04185-DN neurons inhibit rolling.

To further test the role of SS04185-DN neurons, we investigated whether these neurons were involved in reducing the duration of each rolling bout (*Figure 1A, D, F*). As a result, knockdown of SS04185-MB neurons did not increase the duration of rolling bouts (*Figure 2D*). Furthermore, the earlier onset of crawling triggered by the activation of SS04185 neurons remained the same with knockdown of SS04185-MB neurons (*Figure 2E*). Collectively, these results strongly suggest that the behavioral effects on both rolling and crawling, as illustrated in *Figure 1*, are primarily mediated by SS04185-DN neurons.

To further ascertain the role of SS04185-DN neurons in the regulation of rolling, we employed the heat shock FlpOut mosaic expression approach. This technique allowed for controlled and sporadic expression of CsChrimson in SS04185 neurons thorough random induction of Flippase by manipulating the timing and duration of heat shock (*Golic and Lindquist, 1989*; *Nern et al., 2015*). We compared larvae subjected to activation of both SS04185-MB and SS04185-DN neurons (red, *Figure 2—figure supplement 1F*) with those subjected only to activation of SS04185-MB neurons (black, *Figure 2—figure supplement 1G*), to assess the degree to which the former showed behavioral effects. Remarkably, activation of both SS04185-MB and SS04185-DN neurons resulted in a reduction in both the probability and duration of rolling when compared to activation of SS04185-MB neurons alone (*Figure 2F, G*, *Figure 2—figure supplement 1H, I*). Furthermore, activation of both SS04185-MB and

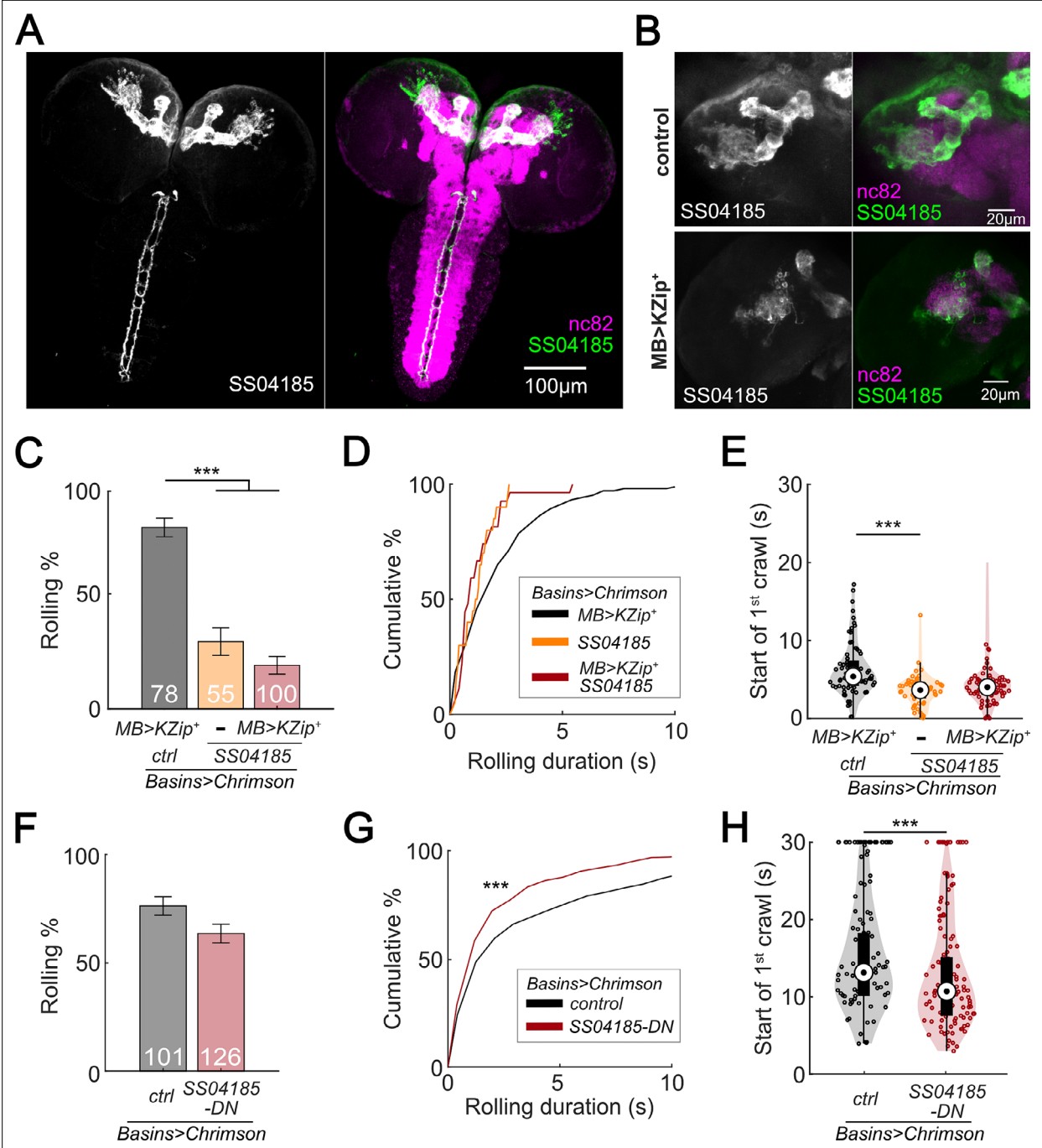

**Figure 2.** SS04185-DN, but not SS04185-MB, inhibits rolling when co-activated with Basins. (**A**) Morphology of SS04185 neurons. GFP, gray (left), green (right); nc82, magenta. Anterior, up; dorsal view; scale bar, 100 µm. Genotype: *10xUAS-IVS-myr::GFP/+; R54B01-Gal4.AD/+; R46E07-Gal4. DBD/+*. (**B**) Kenyon cells are less labeled in SS04185 with MB>Killer Zipper. CsChrimson::mVenus expression in Kenyon cells of SS04185 in Control and SS04185 with Killer Zipper in mushroom body (MB). mVenus, gray (left), green (right); nc82, magenta. Anterior, up; dorsal view; scale bar, 20 µm. Genotype: *20xUAS-IVS-CsChrimson::mVenus/+; R54B01-Gal4.AD/+; R46E07-Gal4.DBD/+* (control); *20xUAS-IVS-CsChrimson::mVenus/+; R13F02-LexA,LexAop-KZip+/R54B01-Gal4.AD; R72F11-Gal4/R46E07-Gal4.DBD* (MB>KZip+). (**C**) Rolling probabilities of larvae with activation of SS04185 reduce the expression of CsChrimson in MB neurons. Error bars, 95% confidence interval. *n* = 78, 55, 100 from left to right. Statistics: Chi-square test, $\chi^2$ = 2.32, p > 0.05 for the two groups with SS04185 expression; $\chi^2$ = 37.50, p < 0.001 for the comparison between the two groups on the left; $\chi^2$ = 70.45, p < 0.001 for the comparison between the groups with *MB>KZip+* expression which reduce expression of CsChrimson in MB. Genotypes: *20xUAS-IVS-CsChrimson::mVenus/+; R13F02-LexA,LexAop-KZip+/+; R72F11-Gal4/+* (black); *20xUAS-IVS-CsChrimson::mVenusR54B01-Gal4.AD/+; R46E07-Gal4. DBD/R72F11-Gal4* (orange); *20xUAS-IVS-CsChrimson::mVenus/+; R13F02-LexA,LexAop-KZip+/R54B01-Gal4.AD; R72F11-Gal4/R46E07-Gal4.DBD* (red). Genotypes in (**D–E**) are the same as mentioned here. (**D**) Cumulative plot of rolling duration. Statistics: Kruskal–Wallis test: *H* = 8.28, p = 0.016;

*Figure 2 continued on next page*

*Figure 2 continued*

Bonferroni-corrected Mann–Whitney $U$ test, p > 0.05 for all pairwise post hoc tests, $n$ = 103, 20, 27 from left to right. (**E**) A violin plot of start of first crawling bout for each larva during stimulation. Statistics: Kruskal–Wallis test: $H$ = 15.02, p < 0.001; Bonferroni-corrected Mann–Whitney $U$ test, p > 0.05 for the two groups with SS04185 expression; p < 0.001 for the comparison between the group without SS04185 expression and the groups with full SS04185 expression, $n$ = 65, 20, 7 from left to right. (**F**) The probabilities of larval rolling during first 5 s of stimulation. Error bars, 95% confidence interval. $n$ = 101, 126. Statistics: Chi-square test, $\chi^2$ = 4.27, p = 0.039. Genotype: *13xLexAop2-IVS-CsChrimson::tdTomato/w⁺, hs-FLP; R54B01-Gal4.AD/72F11-LexA; 20XUAS-(FRT.stop)-CsChrimson::mVenus/R46E07-Gal4.DBD*. Genotypes in (**G, H**) are the same as mentioned here. (**G**) Cumulative plot of rolling duration. Statistics: Mann–Whitney $U$ test, p < 0.001, $n$ = 350, 473. (**H**) A violin plot of start of first crawling bout for each larva during stimulation. Statistics: Mann–Whitney $U$ test, p < 0.001, $n$ = 97, 120. ***p < 0.001.

The online version of this article includes the following source data and figure supplement(s) for figure 2:

**Source data 1.** Excel sheet containing raw data from behavior tracking depicted in *Figure 2D, E, G and H*.

**Source data 2.** Excel sheet containing statistic data from behavior tracking depicted in *Figure 2C-H*.

**Figure supplement 1.** SS04185-DN inhibits rolling.

**Figure supplement 1—source data 1.** Excel sheet containing raw data from behavior tracking depicted in *Figure 2—figure supplement 1C, I and J*.

**Figure supplement 1—source data 2.** Excel sheet containing statistics data from behavior tracking depicted in *Figure 2—figure supplement 1C, E, I and J*.

SS04185-DN neurons reduced the latency to the end of the first rolling bout and the initiation of the first crawling bout (*Figure 2H*, *Figure 2—figure supplement 1J*). These findings provide compelling evidence that SS04185-DN neurons, but not SS04185-MB neurons, play an important role in the termination of rolling. Collectively, the results suggest that a single pair of descending neurons in SS04185 is important for termination of rolling during the activation of Basins.

## Descending neurons identified by SS04185 correspond to SeIN128 neurons

In a previous electron microscopy (EM) connectome study, we identified a set of neurons designated as SeIN128, whose-cell bodies in the SEZ send axonal projections throughout the thoracic and abdominal segments (*Figure 3A*; *Ohyama et al., 2015*). Our immunostaining data also showed that the cell bodies of SS04185-DN neurons are located in the SEZ, with axons bilaterally innervating the medial regions of the ventral nerve cord (VNC) from the thoracic to abdominal segments A8/9 (*Figure 2A*), suggesting that SS04185-DN and SeIN128 neurons are one and the same.

To verify this possibility, we examined the detailed anatomy of SS04185-DN neurons by immunostaining them with several markers and compared our immunostaining images with the corresponding images obtained via EM reconstruction of the entire CNS of a first instar *Drosophila* larva (*Ohyama et al., 2015*; *Winding et al., 2023*). We confirmed that the projections of SeIN128 neurons are distributed within the ventromedial neural tract (one of the six major neural tracts) in *Drosophila* larvae (*Figure 3A–C*) in EM reconstruction data. We also confirmed that the cell bodies of SS04185-DN neurons were again located in the SEZ region, where the most anterior of the three neuropils in the thoracic region was marked by N-cadherin (*Figure 3D*). Viewed from the side (i.e., in the longitudinal or sagittal plane), both the cell bodies and axonal arbor were located ventrally (*Figure 3D*, far right). Immunostaining with Fasciclin2 (Fas2), which labels various neural tracts in the VNC (*Grenningloh et al., 1991*; *Santos et al., 2007*), showed colocalization of the axonal projections of SS04185-DN neurons and the Fas2-labeled ventromedial tract (*Figure 3C, E*). The similarity of the locations of their cell bodies and the distributions of their axonal processes suggests the identity of the SS04185-DN and SeIN128 neurons.

A previous EM study showed that SeIN128 neurons were located downstream of Basin neurons (*Ohyama et al., 2015*). To further confirm the identity of SS04185-DN and SeIN128 neurons, we compared the distributions of the axonal projections of SS04185-DN neurons in relation to those of several key neurons within the rolling circuit: the Basins, A00c neurons (a group of ascending neurons downstream of the Basins, and which facilitate rolling), and mdIV neurons (nociceptive sensory neurons upstream of the Basins). Immunostaining revealed that Basin projections colocalize with those of SS04185-DN neurons in both the horizontal and transverse planes (*Figure 3F*, top and lower panels, respectively), with the horizontal view showing that SS04185-DN projections are distributed slightly medial to those of Basins within the ventromedial tract (*Figure 3F*, top panels), which resembles their

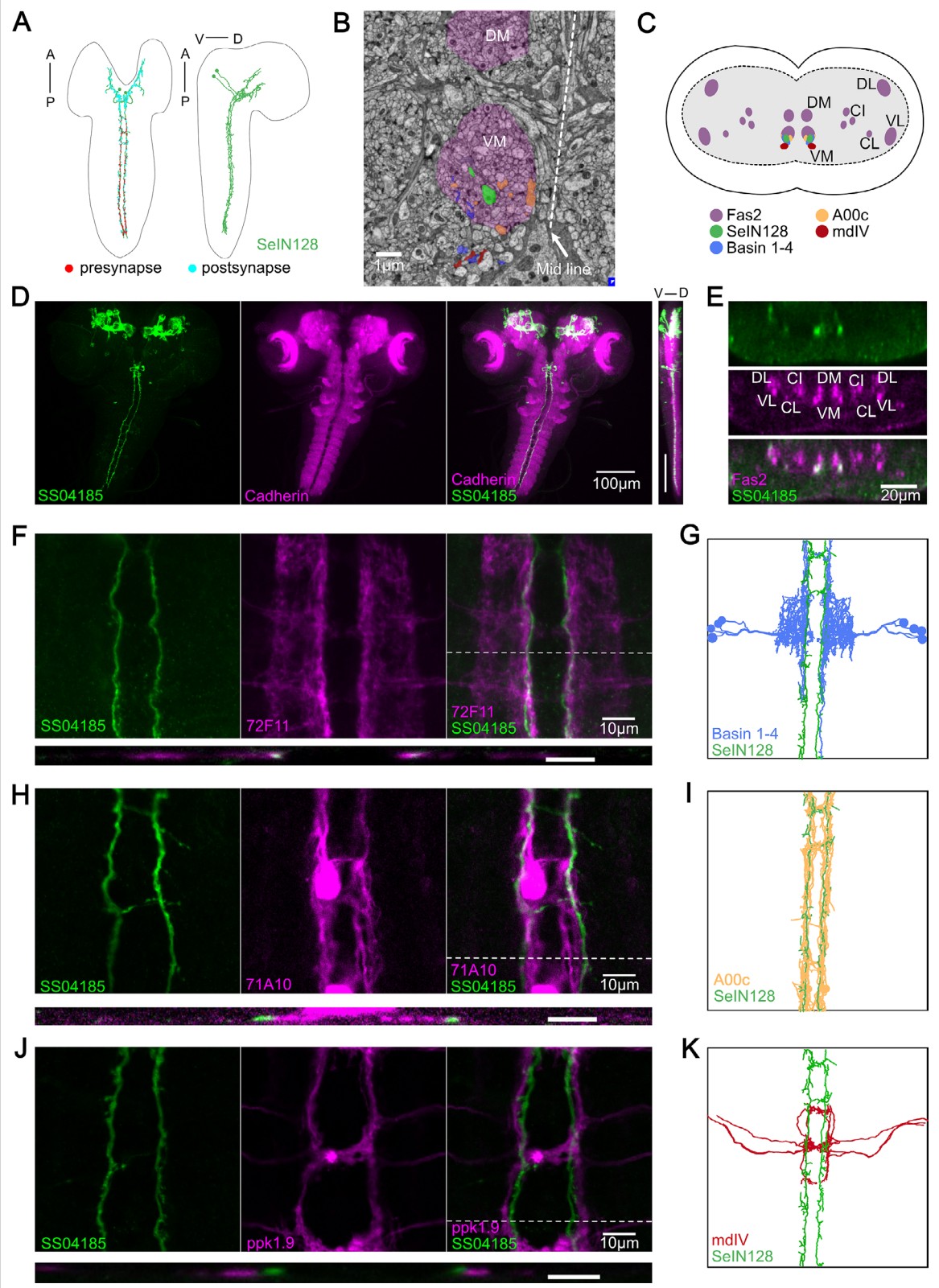

**Figure 3.** SS04185-DN is identical to SeIN128. (**A**) Transmission Electron microscopy (TEM) neuron reconstruction of SeIN128 neurons. Left panel: anterior, up; dorsal view. Right panel: anterior, up; dorsal, right; lateral view. Red dots, presynaptic sites. Cyan dots, postsynaptic sites. (**B**) A transverse section of larval central nervous system (CNS) from EM reconstruction data. SeIN128 (green), Basins (blue), and A00c (orange) are located in ventromedial tract (VM). mdIV, red; magenta, neural tracts. DM, dorsomedial tract; VM, ventromedial tract. Dorsal, up; anterior view; scale bar, 1 µm.

*Figure 3 continued on next page*

*Figure 3 continued*

(**C**) Cartoon generated based on transverse section of SeIN128, Basin-1 to Basin-4, A00c, and mdIV from EM neuron reconstruction data and (**D**). Nerve tracts are shown in magenta. Dorsal, up; posterior view. DM, dorsomedial tract; VM, ventromedial tract; CI, central-intermediate tract; CL, central-lateral tract; DL, dorsolateral tract; VL, ventrolateral tract. SeIN128, green; Basin-1 to Basin-4, blue; A00c, orange; mdIV, red. (**D**) SS04185-expressing neurons co-stained with N-cadherin. A cell body of SS04185-descending neuron located in ventral part of the subesophageal zone (SEZ). SS04185, green; N-cadherin, magenta. Anterior, up; left, dorsal view; right, longitudinal section; scale bar, 100 µm. Genotype: *10xUAS-IVS-myr::GFP/+; R54B01-Gal4.AD/+; R46E07-Gal4.DBD/+*. SS04185, green; Cadherin, magenta. Anterior, left, dorsal, up; lateral view; scale bar, 100 µm. (**E**) Transverse section of SS04185-DN co-stained with Fas2. SS04185-DN located at ventromedial tract (VM). SS04185, green; Fas2, magenta. Dorsal, up; posterior view; scale bar, 20 µm. DM, dorsomedial tract; VM, ventromedial tract; CI, central-intermediate tract; CL, central-lateral tract; DL, dorsolateral tract; VL, ventrolateral tract. Genotype: *10xUAS-IVS-myr::GFP/+; R54B01-Gal4.AD/+; R46E07-Gal4.DBD/+*. (**F, H, J**) SS04185-DN co-localized with Basins or A00C neuron tract but not MdIV. SS04185, green; Basins (**F**), A00c (**H**), or mdIV (**J**), magenta. Genotype: *w; R54B01-Gal4.AD/R72F11-LexA(F) 71A10-LexA(H) or ppk1.9-LexA(J); R46E07-Gal4.DBD/13xLexAop2-IVS-CsChrimson::tdTomato,20xUAS-IVS-GCaMP6s*. Top panel: anterior, up; dorsal view; scale bar, 10 µm. Bottom panel: dorsal, up; posterior view; scale bar, 5 µm. (**G, I, K**) SeIN128, Basin-2, A00c, or mdIV morphologies from the TEM neural reconstruction. Anterior, up; dorsal view. SS04185, green; Basin-2, blue; A00C, orange; mdIV, red.

colocalization pattern reported in EM (*Figure 3B, C, G*). Similarly, we compared the distributions of SS04185-DN projections with those of A00c or mdIV projections. We found that the projections of A00c colocalize with those of SS04185-DN in a similar fashion along the rostrocaudal axis within the ventromedial tract (*Figure 3H,I*), with A00c projections distributed more medially than SS04185-DN projections, consistent with the distribution patterns of SeIN128 projections and A00c projections in the EM reconstruction dataset (*Figure 3B, H, I*). In contrast, the distributions of mdIV projections did not colocalize with those of SS04185-DN projections, as the mdIV projections were displaced more laterally relative to the SS04185-DN projections in the horizontal and transverse planes (*Figure 3J*, top and lower panels, respectively), consistent with the distribution patterns of SeIN128 and mdIV projections in the EM reconstruction dataset (*Figure 3K*). In the transverse plane, the projections of SS04185-DN neurons were also distributed dorsomedial to those of mdIV (*Figure 3J*, lower panel), consistent with the corresponding distribution patterns in the EM reconstruction dataset (*Figure 3B, C, K*).

We conclude that the morphological findings for SS04185-DN neurons, together with data on the distribution of their axonal projections in relation to that of Basin, A00c, and mdIV neurons, strongly suggest the identity of SS04185-DN and SeIN128 neurons.

## Connectome and functional connectivity analyses: SeIN128 neurons receive inputs from Basin-2 and A00c

A previous study that reconstructed larval neurons involved in the rolling circuit showed that Basin-2 and A00c neurons (in the VNC) make excitatory synaptic contacts onto SeIN128 neurons (in the CNS), which in turn make reciprocal inhibitory synaptic contacts onto Basin-2 and A00c neurons (*Figure 4A*, *Figure 4—figure supplement 1A*; *Ohyama et al., 2015*). These data suggest that SeIN128 neurons are directly activated by Basin-2 and A00c (which also receives inputs from Basin-1, Basin-2, and Basin-4).

To assess the functional significance of these synaptic connections between SeIN128 neurons and Basins or A00c, we activated either Basins or A00c neurons and examined the resultant green fluorescent protein (GFP) calmodulin (CaM) protein (GCaMP) signaling in SeIN128 neurons. Specifically, after expressing CsChrimson in Basins and A00c neurons and GCaMP in SeIN128 neurons, we used a two-photon microscope (920 nm laser) and monitored GCaMP signaling in SeIN128 neurons during illumination of a specimen with a 620-nm light-emitting diode (LED) for 1 s (0.04–1.4 µW/mm²), which activated either Basins or A00c neurons. GCaMP signals in SeIN128 neurons increased in an intensity-dependent manner when either Basins and A00c were activated (*Figure 4B, C*) but not when larvae were not fed retinal (*Figure 4—figure supplement 1C, D*). Peak activity occurred at around 3 s after the onset of LED stimulation, which was similar to the results when Basins or A00c neurons were stimulated (*Figure 4B, C*). Finally, both Basin and A00c stimulation resulted in linear dose-dependent increases in SeIN128 firing (*Figure 4—figure supplement 1B*). These results are consistent with the notion that SeIN128 neurons are downstream of Basins and A00c neurons.

To compare the neural responses between Basins and SeIN128 or A00c neurons, we recorded neural activity in A00c neurons with GCaMP while stimulating Basin neurons in the same experimental setting. Although A00c neurons displayed a similar dose-dependent increase in peak axonal firing

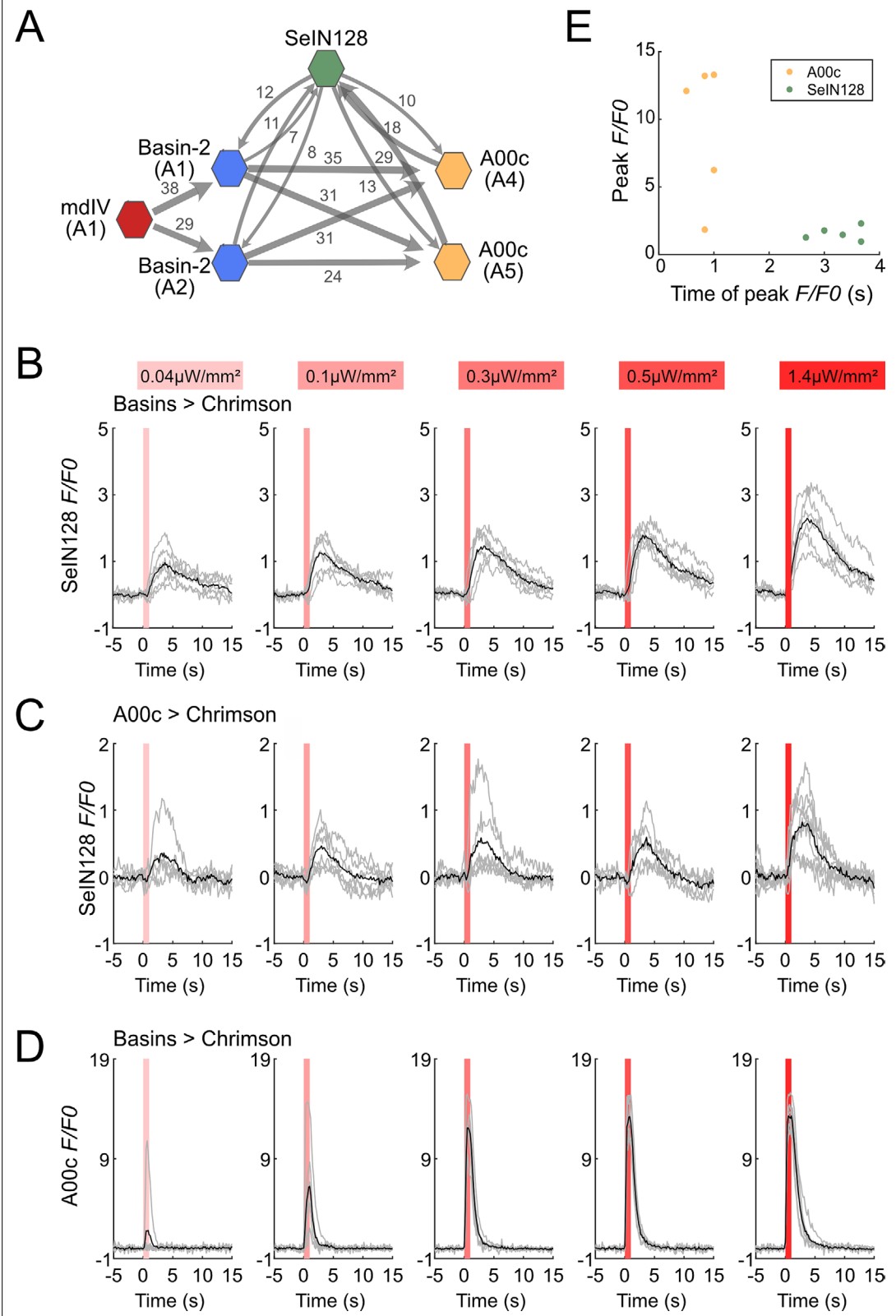

**Figure 4.** SeIN128 receives input from Basin and A00c neurons. (**A**) Summary of the connectivity between SeIN128 and the escape circuit. SeIN128 receives inputs from Basin-2 and A00c and provide feedback to Basin-2 and A00c. Synapse number shown next to connection arrows, where line width is proportional to synapse number. All connections in the ventral nerve cord are shown except unilateral synapses, <5 synapses, between neurons. Each polygon represents a pair of the indicated neuron and segment (segment number is shown under the neuron name). SeIN128, green; Basin-2, blue;

*Figure 4 continued on next page*

*Figure 4 continued*

A00c, orange; mdIV, red. SeIN128 is functionally downstream of Basins (**B**) or A00c (**C**). Calcium transients, $\Delta F/F_0$ traces of GCaMP6s in SeIN128 axons (black line, mean; gray line, single larva) during 610 nm optogenetic activation of Basins at various intensities. Vertical gray line represents optogenetic activation. Genotype: *w; R72F11-LexA* (**B**) *or R71A10-LexA* (**C**) */R54B01-Gal4.AD; 13xLexAop2-IVS—CsChrimson::tdTomato, 20xUAS-IVS-GCaMP6s/ R46E07-Gal4.DBD.* (**D**) A00c responses are faster and stronger than SeIN128 responses during activation of Basins. Calcium transients (black line, mean; gray line, single larva) represented by $\Delta F/F_0$ in A00c by of 610 nm optogenetic activation of Basins at various intensities. Genotype: *w; R72F11-LexA/+; 13xLexAop2-IVS-CsChrimson::tdTomato, 20xUAS-IVS-GCaMP6s/R71A10-Gal4.* For (**B–D**), irradiances from left to right are 0.04, 0.1, 0.3, 0.5, and 1.4 µW/mm². For each irradiance ($n = 6$), individual traces are shown with gray lines whereas the average of individuals is shown in black. The shaded gray area indicates the period of optogenetic activation (0–1 s). (**E**) The timing of the peak $\Delta F/F_0$ correlated with the identity of the neurons but not the peak $\Delta F/F_0$ value. SeIN128 neurons are shown as orange dots, whereas A00c is shown as a green dot.

The online version of this article includes the following source data and figure supplement(s) for figure 4:

**Source data 1.** Excel sheet containing raw data depicted in *Figure 4C*.

**Figure supplement 1.** SeIN128 is downstream of Basin and A00c neurons.

**Figure supplement 1—source data 1.** Excel sheet containing raw data depicted in *Figure 4A and B*.

**Figure supplement 2.** Synapses from SeIN128 to Basin-2 are located near Basin-2 outputs.

as the intensity of optogenetic stimulation of Basin neurons increased, unlike SeIN128 neurons they showed no delay in peak firing activity (*Figure 4D, E*, *Figure 4—figure supplement 1B*), suggesting that A00c and SeIN128 neurons function differently in the rolling circuit.

We then investigated the anatomical locations of the synaptic outputs and inputs of SeIN128 neurons, and found that, whereas their outgoing projections primarily make synaptic contacts along the anterior–posterior nerve axis, the inputs coming from other neurons are mainly located in the SEZ (*Figure 3A*). On the other hand, SeIN128 neurons make axo-axonal contacts onto Basin-2 neurons (*Figure 4—figure supplement 2A–G*) as well as A00c neurons: that is, their axons make synaptic contacts with the dorsal and medial processes of Basin-2, which correspond to their axonal compartments (*Figure 4—figure supplement 2E–G*). These data suggest that the delay of SeIN128 activity may be caused by multi-synaptic connections involving the SEZ or a feedback loop involving axo-axonal connections between SeIN128 and Basin-2 or A00c.

## SeIN128 neurons are GABAergic and inhibitory

The results thus far indicate that, activation of SeIN128 neurons inhibits rolling (*Figure 1A–C*); SeIN128 neurons receive functional inputs from Basin-2 and A00c (*Figure 4A–C*); and SeIN128 neurons make anatomical connections onto Basin-2 and A00c (*Figure 4A*). These findings suggest that SeIN128 neurons might be inhibitory. To test this possibility, we performed immunostaining experiments and found that SeIN128 neurons colocalized with glutamic acid decarboxylase (Gad)-positive neurons but not with acetylcholine- or glutamate-positive neurons, suggesting that SeIN128 neurons are GABAergic inhibitory neurons (*Figure 5A*, *Figure 5—figure supplement 1A,B*).

We reasoned that if Gamma-aminobutyric acid (GABA) in SeIN128 neurons is necessary for inhibiting rolling, then selectively knocking down GABA secretion in SeIN128 neurons should enhance rolling. When we expressed RNA interference (RNAi) *HMS02355* in SeIN128 neurons to knock down vesicular GABA transporter (VGAT) expression and suppress the release of GABA, the population-level rolling probability increased from 23.6% to 45.2% (*Figure 5B, C*; *Kallman et al., 2015*; *Zhao et al., 2019*). We confirmed the effect of *HMS02355* by immunostaining: pan-neural *HMS02355* expression decreased GABA and VGAT expression in the neuropil (*Figure 5—figure supplement 1C, D*). The control group (only Basins expressing CsChrimson with VGAT RNAi HMS02355 but without SS04185) showed a lower probability of rolling (23.6%) compared to similar genotypes without VGAT RNAi HMS02355 (*Figure 2C, F*). This indicates that VGAT RNAi HMS02355 background reduces the probability of rolling. Furthermore, the duration of each bout of rolling increased from 0.8 to 1.4 s (*Figure 5D*). These data support the idea that SeIN128 neurons inhibit rolling via GABAergic transmission.

## Inhibition of SeIN128 increases probability and duration of rolling

To further test whether the release of GABA upon activating SeIN128 neurons is necessary for inhibiting rolling, we expressed tetanus toxin (TNT) in SeIN128 neurons to block synaptic transmission.

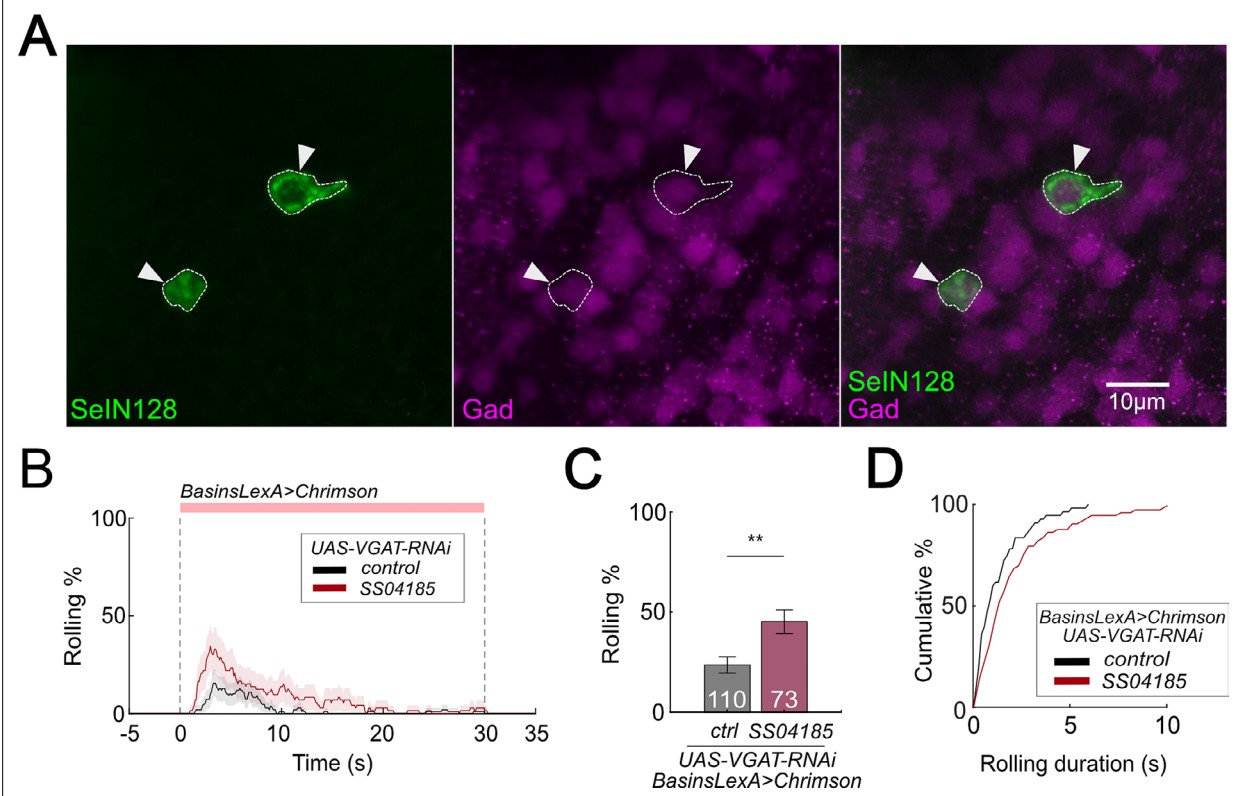

**Figure 5.** SeIN128 is GABAergic and negatively controls rolling. (**A**) Immunostaining of SeIN128 cell body (green) and GABAergic neuron (magenta). Genotype: *10xUAS-IVS-myr::GFP; R54B01-Gal4.AD/13xLexAop-dsRed; R46E07-Gal4.DBD/Trojan-GAD-T2A-LexA*. White triangles indicate locations of SeIN128 cell bodies. Anterior, up; dorsal view; scale bar, 10 μm. (**B**) Time series of rolling probabilities of larvae with Basin activation (black), or vesicular GABA transporter (VGAT) RNA interference (RNAi) in SS04185 and Basin activation (red). The red bar and dashed lines display the window of optogenetic stimulation eliciting larval escape responses. Shaded areas show 95% confidential intervals of rolling probabilities. Genotypes: *13xLexAop2-IVS-CsChrimson::mVenus; R72F11-LexA/+; HMS02355/+* (black); *13xLexAop2-IVS-CsChrimson::mVenus; R72F11-LexA/R54B01-Gal4. AD; HMS02355/R46E07-Gal4.DBD* (red). Genotypes in (**C, D**) are the same as mentioned here. (**C**) Binned larval rolling probabilities during first 5 s of stimulation in (**A**). Error bars, 95% confidence interval. *n* = 110, 73. Statistics: Chi-square test, $\chi^2$ = 9.34, p < 0.001. (**D**) Cumulative plot of rolling duration. Statistics: Mann–Whitney *U* test, p = 0.015, *n* = 55, 73. **p < 0.01.

The online version of this article includes the following source data and figure supplement(s) for figure 5:

**Source data 1.** Excel sheet containing raw data depicted in *Figure 5D*.

**Source data 2.** Excel sheet containing statistics data depicted in *Figure 5C and D*.

**Figure supplement 1.** SeIN128 is GABAergic.

**Figure supplement 1—source data 1.** Excel sheet containing raw data depicted in *Figure 5—figure supplement 1C and D*.

**Figure supplement 1—source data 2.** Excel sheet containing statistics data depicted in *Figure 5—figure supplement 1C and D*.

Silencing SeIN128 neurons via TNT while triggering rolling by optogenetically activating Basin neurons via *R72F11-LexA>LexAop-CsChrimson* significantly increased the probability of rolling compared to controls (*Figure 6A, B*). Silencing SeIN128 neurons via TNT extended the duration of each rolling bout, as well as the total rolling duration, in each larva (*Figure 6C, D*). We also examined the rolling-escape crawling sequence upon silencing SeIN128 neurons, and found that the time to offset of rolling and the time onset of crawling were both delayed relative to controls (*Figure 6E, F*).

Given that TNT is expressed constitutively during development, long-term compensatory changes in the nervous system could have contributed to alterations in the parameters of rolling and crawling. To test whether similar results could be replicated with the use of a temporally specific intervention, we expressed *shibire*[ts1] (*shi*[ts1]) in SeIN128 neurons to block synaptic transmission at temperatures above 30°C (*van de Goor et al., 1995*; *Kitamoto, 2001*). Silencing SeIN128 neurons via *shibire*[ts1] increased the probability of rolling from 60.4% to 79.7% (*Figure 6—figure supplement 1A, B*). The total duration of rolling per animal during stimulation increased from 10 to 12 s (*Figure 6—figure*

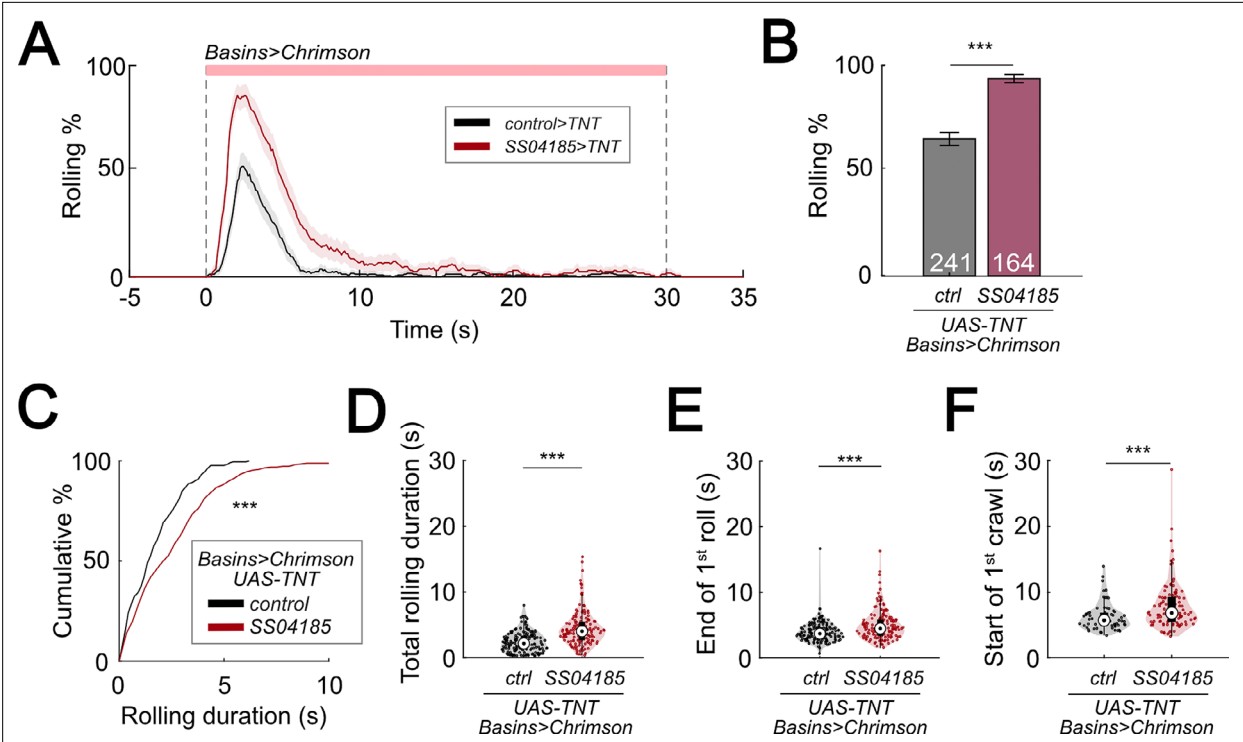

**Figure 6.** Inhibition of SeIN128 prolongs rolling and delays initiation of crawling. (**A**) Time series of rolling probabilities of larvae with Basin activation (black), or SS04185 inhibition and Basin activation (red). Shaded regions show 95% confidential intervals of rolling probabilities. Genotypes: *13xLexAop2-IVS-CsChrimson::mVenus; R72F11-LexA/+; UAS- TeTxLC.tnt/+* (black); *13xLexAop2-IVS-CsChrimson::mVenus; R72F11-LexA/R54B01-Gal4.AD; UAS-TeTxLC.tnt/R46E07-Gal4.DBD* (red). Genotypes in (**B–F**) are the same as mentioned here. (**B**) Rolling probabilities during first 5 s of stimulation in (**A**). Error bars, 95% confidence interval. $n = 241, 164$. Statistics: Chi-square test, $\chi^2 = 44.02$, $p < 0.001$. (**C**) A violin plot of total time spent rolling for each individual larva during stimulation. Statistics: Mann–Whitney $U$ test, $p < 0.001$, $n = 221, 258$. (**D**) Cumulative plot of rolling duration. Statistics: Mann–Whitney $U$ test, $p < 0.001$, $n = 160, 154$. (**E**) A violin plot of end of first rolling bout for each larva during stimulation. Statistics: Mann–Whitney $U$ test, $p < 0.001$, $n = 160, 154$. (**F**) A violin plot of start of first crawling bout for each larva during stimulation. Statistics: Mann–Whitney $U$ test, $p < 0.001$, $n = 65, 105$. ***$p < 0.001$.

The online version of this article includes the following source data and figure supplement(s) for figure 6:

**Source data 1.** Excel sheet containing raw data from behavior tracking depicted in *Figure 6C-F*.

**Source data 2.** Excel sheet containing statistics data from behavior tracking depicted in *Figure 6C-F*.

**Figure supplement 1.** SeIN128 inhibition enhances rolling.

supplement 1C). Although the duration of each rolling bout, the time to onset of the first rolling bout, and time to onset of the first crawling bout did not differ from those of controls (*Figure 6—figure supplement 1D, E, G*), the time to offset of the first rolling bout was delayed relative to controls ($p = 0.013$ for *Figure 6—figure supplement 1F*). Together with the results showing that activation of SeIN128 neurons inhibits rolling, these findings suggest that the activity of SeIN128 neurons is important in controlling the duration of rolling and the shift to crawling.

## Basins receive GABAergic inputs that inhibit rolling

Given that Basins receive axo-axonal inputs from SeIN128 neurons and GABA signaling in SeIN128 neurons inhibits rolling, we next used RNAi to test whether Basins receive GABAergic signals from SeIN128. We hypothesized that knockdown of GABA receptors in Basin neurons would increase the probability and duration of rolling at the population level. To knock down ionotropic GABA-A receptors (GABA-A-R) and G-protein-coupled GABA-B receptors (GABA-B-R1 and GABA-B-R2), we tested Basin neurons with GABA-A-R, GABA-B-R1, and GABA-B-R2 RNAi lines (i.e., *HMC03643* for GABA-A-R, *HMC03388* for GABA-B-R1[1], *JF02989* for GABA-B-R1[2], and *HMC02975* for GABA-B-R2, respectively). For all RNAi lines, the rolling probability at the population level increased from 80% to 90% or even higher (*Figure 7A*), while the total rolling duration at the individual level increased for

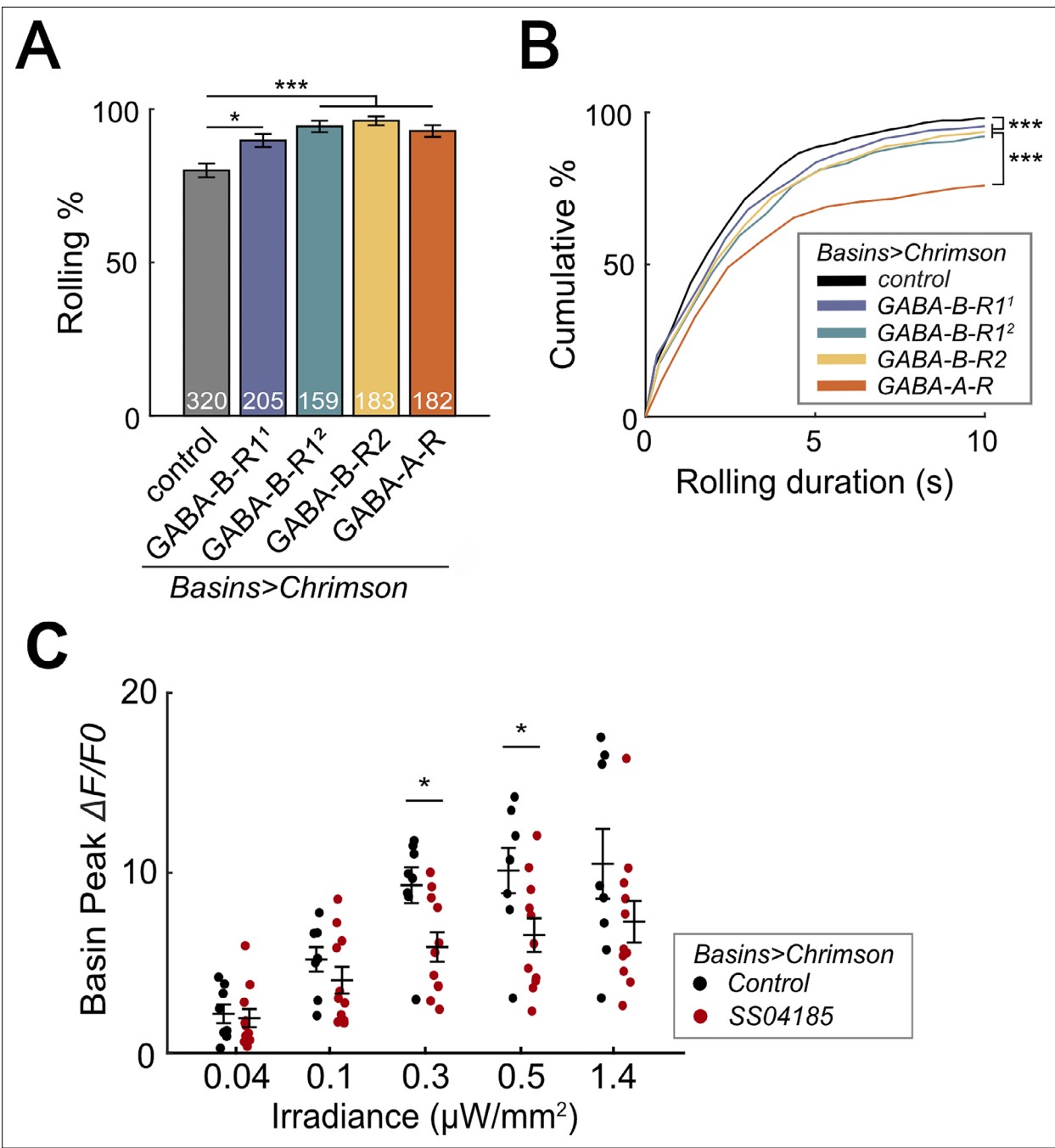

**Figure 7.** SeIN128 sends feedback inhibition to Basins. (**A**) Rolling probabilities for larvae with GABAR-RNAi in their Basin neurons. From left to right, the genotypes are *20xUAS-IVS-CsChrimson::mVenus/+; R72F11-Gal4/+* (black), *20xUAS—IVS-CsChrimson::mVenus/+; R72F11-Gal4/UAS-HMC03388* (blue), *20xUAS-IVS-CsChrimson::mVenus/+; R72F11-Gal4/UAS-JF02989* (green), *20xUAS-IVS-CsChrimson::mVenus/+; R72F11-Gal4/UAS-HMC02975* (yellow), and *20xUAS-IVS-CsChrimson::mVenus/+; R72F11-Gal4/UAS-HMC03643* (orange). Genotypes in (**B**) are the same as mentioned here. N = 320, 205, 159, 183, 182 from left to right. Statistics: Chi-square test, Bonferroni correction. GABA-B-R1[1] group: $\chi^2 = 8.76$, p = 0.012. GABA-B-R1[2] group: $\chi^2 = 24.70$, p < 0.001. GABA-B-R2 group: $\chi^2 = 25.77$, p < 0.001. GABA-A-R group: $\chi^2 = 16.29$, p < 0.001. (**B**) Cumulative plot of rolling duration. Statistics: Kruskal–Wallis test: H = 69.52, p < 0.001; Bonferroni-corrected Mann–Whitney *U* test, p < 0.001 between control and the GABA-B-R11, GABA-B-R12, and GABA-B-R2 RNA interference (RNAi) groups; p < 0.001 between GABA-A-R and all other RNAi groups. Sample sizes for the colored bars from top (control, black) to bottom (GABA-A-R, red); n = 520, 488, 387, 582, 306. (**C**) Summary of peak $\Delta F/F_0$ in Basin axons with or without SeIN128 activation under various irradiances. Control groups shown in black are without SeIN128 activation while experimental groups shown in red are with SeIN128 activation. Statistics: Mann–Whitney *U* test, p > 0.05 for irradiances of 0.04, 0.1, and 1.4 μW/mm²; p = 0.016 for irradiance of 0.3 μW/mm²; p = 0.032 for irradiance of 0.5 μW/mm². Genotype: *20xUAS-Syn21-opGCaMP6s, 10xUAS-Syn21-CsChrimson88::tdTomato/+; CyO/+;R72F11-Gal4/TM6* (black);

*Figure 7 continued on next page*

Figure 7 continued

*20xUAS-Syn21-opGCaMP6s,10xUAS-Syn21-CsChrimson88::tdTomato/+;CyO/R54B01-Gal4.AD;R72F11-Gal4/R46E07-Gal4.DBD* (red). *p < 0.05, ***p < 0.001.

The online version of this article includes the following source data and figure supplement(s) for figure 7:

**Source data 1.** Excel sheet containing raw data from behavior tracking depicted in *Figure 7B and C*.

**Source data 2.** Excel sheet containing statistics data from behavior tracking depicted in *Figure 7B and C*.

**Figure supplement 1.** SeIN128 sends feedback inhibition to Basins.

**Figure supplement 1—source data 1.** Excel sheet containing raw data from behavior tracking depicted in *Figure 7—figure supplement 1*.

**Figure supplement 1—source data 2.** Excel sheet containing statistics data from behavior tracking depicted in *Figure 7—figure supplement 1*.

each larva throughout the stimulation window (*Figure 7—figure supplement 1A*). All GABA receptor knockdown groups showed significant increases in rolling duration across multiple bouts (*Figure 7B*); all groups except for GABA-B-R1[1] showed a reduced time to onset of the first rolling bout (*Figure 7—figure supplement 1B*); and only the GABA-B-R2 and GABA-A-R groups showed a delayed offset of the first rolling bout (*Figure 7—figure supplement 1C*). None of the groups differed from controls in the time to onset of the first crawling bout (*Figure 7—figure supplement 1D*). The greatest increase in the probability and duration of rolling was seen during knockdown of ionotropic GABA-A-R (*Rdl*), suggesting that Rdl contributes most to the inhibition of Basin neurons (*Figure 7A, B*).

To investigate whether SeIN128 neurons actually inhibit Basins, we recorded the activity of all Basins during activation of SeIN128 neurons. We compared GCaMP signaling in the Basins when they were co-activated with SeIN128 neurons (experimental treatment) or when they were activated alone (control treatment), with the intensity of optogenetic stimulation varied from 0.04 to 1.4 µW/mm$^2$. We found that Basins in the experimental group showed reductions in GCaMP signaling by 11–36% compared to those in the control group (*Figure 7C*, *Figure 7—figure supplement 1E, F*). The reductions were observed at all stimulation intensities when contrasting peak GCaMP responses, and statistically significant at intensities of 0.3 and 0.5 µW/mm$^2$ (*Figure 7C*, *Figure 7—figure supplement 1E, F*). Collectively, these data support the idea that SeIN128 neurons directly inhibit the activity of Basins via GABA.

## Effects of SeIN128 activation on rolling elicited by activating individual Basins

In the studies above, we measured the activity of all Basins while manipulating the activity of SeIN128 neurons. Connectome and behavioral analyses indicate, however, that of the four types of Basins, only Basin-2 and Basin-4 receive nociceptive input from mdIV and trigger rolling (*Ohyama et al., 2015*). Moreover, as noted above, an examination of the larval connectome (*Ohyama et al., 2015*; *Winding et al., 2023*) revealed that Basin-2 both receives axo-axonal inputs from SeIN128 neurons and sends excitatory projections to the same SeIN128 neurons, whereas a similar examination revealed that Basin-4 neither receives inputs from, nor sends any outputs to, SeIN128 neurons. Therefore, we hypothesized that activation of SeIN128 neurons would inhibit rolling elicited by Basin-2 activation and modify the temporal parameters of rolling, but not affect rolling elicited by Basin-4 activation.

We first examined the pattern of rolling evoked by optogenetically activating Basin-2. Basin-2 activation induced multiple bouts of rolling throughout the stimulation window (*Figure 8—figure supplement 1A*). Furthermore, the rolling elicited by Basin-2 activation tended to be sustained (*Figure 8—figure supplement 1A*). Next, to determine how SeIN128 activation affects the pattern of rolling elicited by Basin-2 activation, we optogenetically activated SeIN128 neurons and Basin-2 simultaneously. As expected, compared to the probability of rolling in control animals in which only Basin-2 was activated, the probability of rolling in experimental animals in which Basin-2 and SeIN128 neurons were simultaneously activated was significantly lower (66.7% vs 24.4%; *Figure 8A*, *Figure 8—figure supplement 1D*). We also examined other parameters of rolling, including the time from the start (onset) of stimulation to the onset of the first rolling bout, termination (offset) of the first rolling bout, and onset of the first crawling bout, as well as the duration of the rolling bout (i.e., the time from its onset to its offset). Consistent with the hypothesis that SeIN128 activation inhibits Basin-2 activity, the duration of the rolling bout significantly decreased (*Figure 8B*, Mann–Whitney *U* test, p = 0.0034, Cohen's *d* = 0.351) and the time to onset of the first rolling bout significantly increased in experimental

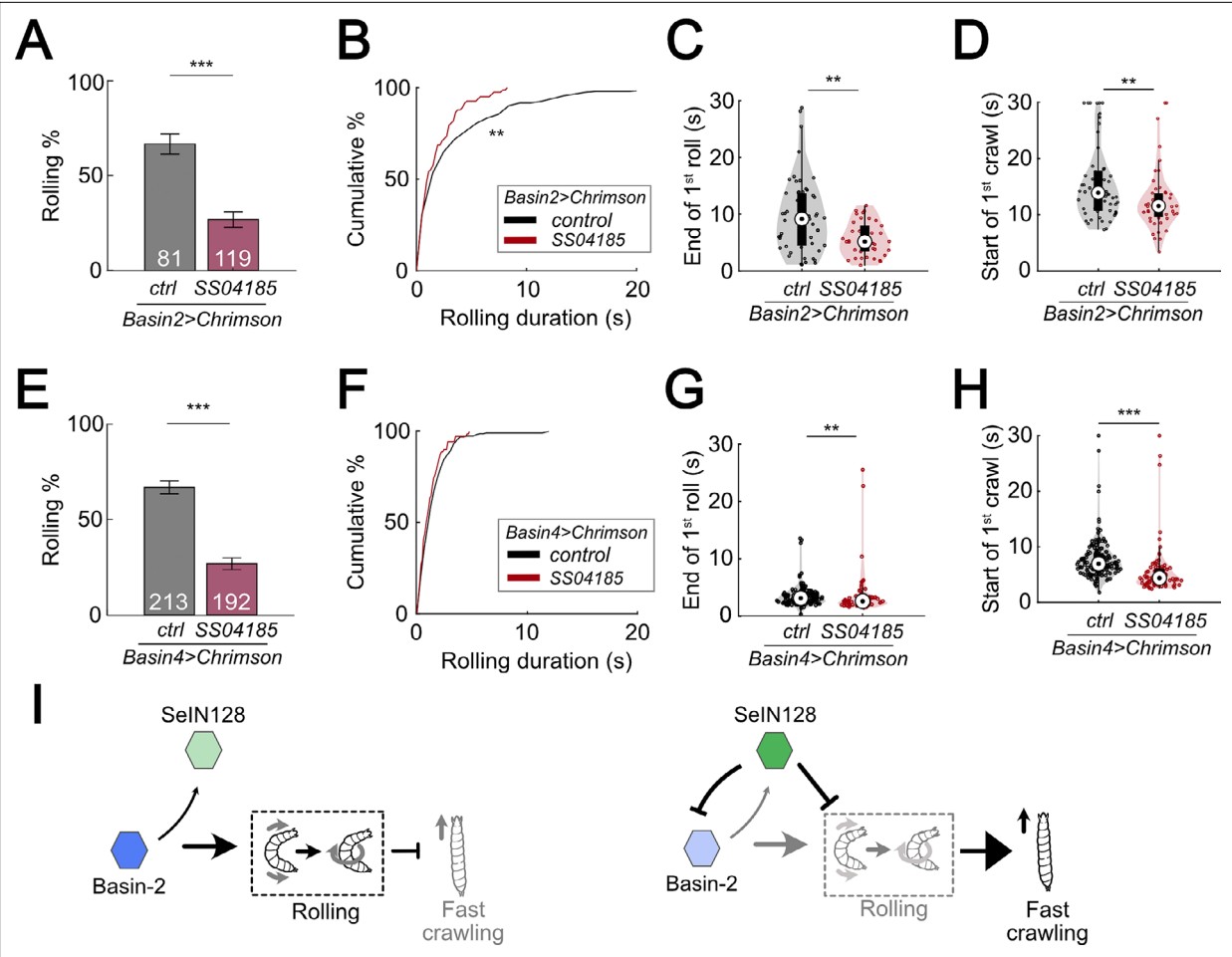

**Figure 8.** SeIN128 inhibits rolling elicited by both Basin-2 and Basin-4 activation. (**A**) Binned larval rolling probabilities during the first 5 s of stimulation. Error bars, 95% confidence interval. $n$ = 81, 119. Statistics: Chi-square test, $\chi^2$ = 35.51, p < 0.001. Genotypes: *20xUAS-IVS-CsChrimson::mVenus/+; R72F11-Gal4.AD/+; R38H09-Gal4.DBD/+* (black); *20xUAS-IVS-CsChrimson::mVenus/+; R72F11-Gal4.AD/R54B01-Gal4.AD; R38H09-Gal4.DBD/R46E07-Gal4.DBD* (red). Genotypes in (**B–D**) are the same as mentioned here. (**B**) Cumulative plot of rolling duration. Statistics: Mann–Whitney *U* test, p = 0.0034, $n$ = 206, 83. (**C**) A violin plot of end of first rolling bout for each larva during stimulation. Statistics: Mann–Whitney *U* test, p = 0.0047, $n$ = 57, 38. (**D**) A violin plot of start of first crawling bout for each larva during stimulation. Statistics: Mann–Whitney *U* test, p = 0.045, $n$ = 107, 38. (**E**) Binned larval rolling probabilities during first 5 s of stimulation. Error bars, 95% confidence interval. $n$ = 192, 213. Statistics: Chi-square test, $\chi^2$ = 64.81, p < 0.001. Genotypes: *20xUAS-IVS-CsChrimson::mVenus/+; R72F11-Gal4.AD/+; R57F07-Gal4.DBD/+* (black); *20xUAS-IVS-CsChrimson::mVenus/+; R72F11-Gal4. AD/R54B01-Gal4.AD; R57F07-Gal4.DBD/R46E07-Gal4.DBD* (red). Genotypes in (**F–H**) are the same as mentioned here. (**F**) Cumulative plot of rolling duration. Statistics: Mann–Whitney *U* test, p = 0.032, $n$ = 231, 71. (**G**) A violin plot of end of first rolling bout for each larva during stimulation. Statistics: Mann–Whitney *U* test, p = 0.0047, $n$ = 129, 61. (**H**) A violin plot of start of first crawling bout for each larva during stimulation. Statistics: Mann–Whitney *U* test, p < 0.001, $n$ = 159, 71. (**I**) A summarizing illustration. Basin-2 activates rolling and supresses fast crawling, while SeIN128 decreases Basin-2 activities to inhibit rolling and disinhibit fast crawling. Arrows show activation and blunt ends represent inhibition. **p < 0.01, ***p < 0.001.

The online version of this article includes the following source data and figure supplement(s) for figure 8:

**Source data 1.** Excel sheet containing raw data from behavior tracking depicted in *Figure 8B-D and F-H*.

**Source data 2.** Excel sheet containing statistics data from behavior tracking depicted in *Figure 8B-D and F-H*.

**Figure supplement 1.** SeIN128 inhibits rolling elicited by both Basin-2 and Basin-4 activation.

**Figure supplement 1—source data 1.** Excel sheet containing raw data from behavior tracking depicted in *Figure 8—figure supplement 1C, E and G*.

**Figure supplement 1—source data 2.** Excel sheet containing statistics data from behavior tracking depicted in *Figure 8—figure supplement 1C, E and G*.

animals compared to controls (*Figure 8—figure supplement 1E*; Mann–Whitney *U* test, p < 0.001). In addition, as expected, the time to offset of the first rolling bout (*Figure 8C*; Mann–Whitney *U* test, p = 0.0047, Cohen's *d* = 0.607) and time to onset of the first crawling bout (*Figure 8D*; Mann–Whitney *U* test, p = 0.0074, Cohen's *d* = 0.548) both significantly decreased in experimental animals compared to controls. Collectively, these findings suggest that Basin-2 neurons play a major role in mediating the effects of SeIN128 activation on rolling induced by optogenetic activation of all Basin neurons.

To ascertain our expectation that SeIN128 activation would have little if any effect on the pattern of rolling elicited by Basin-4 activation, given the absence of any identifiable synaptic contacts between Basin-4 neurons and SeIN128 neurons based on available information on the larval connectome, we also carried out the same analyses as those described above for rolling elicited by Basin-2 activation. We examined the pattern of rolling evoked by optogenetically activating Basin-4, and found that this manipulation induced rolling mostly within the first 5 s of stimulation (*Figure 8—figure supplement 1B, F*). Consequently, at the population level, rolling elicited by Basin-4 activation was transient compared to the rolling elicited by Basin-2 activation (compare *Figure 8—figure supplement 1A, B*; *Figure 8—figure supplement 1C*).

We then assessed whether SeIN128 activation would affect rolling elicited by Basin-4 activation. Surprisingly, compared to control animals, the probability of rolling in experimental animals was significantly lower (66.7% vs 26.8%; *Figure 8E*), much as was the case for rolling elicited by Basin-2 activation. We also examined the other rolling parameters, and found that the duration of the rolling bouts (*Figure 8F*; Mann–Whitney *U* test, p = 0.032, Cohen's *d* = 0.248), time to offset of the first rolling bout (*Figure 8G*; Mann–Whitney *U* test, p < 0.0047, Cohen's *d* = 0.427), and time to onset of the first crawling bout (*Figure 8H*; Mann–Whitney *U* test, p < 0.001, Cohen's *d* = 1.039) all significantly decreased in experimental animals compared to controls, although the effect sizes were smaller compared to those observed for rolling elicited by Basin-2 activation. The time to onset of the first rolling bout, however, did not significantly differ between experimental animals and controls (*Figure 8—figure supplement 1G*). These findings suggest the possibility that sites further downstream of Basin-4 neurons may be involved in inhibitory processes that affect rolling elicited by Basin-4 activation.

## Discussion

In this study, we provide both anatomical and functional evidence that, bilateral descending neurons in the brain of a *D. melanogaster* larva, which comprise part of a neural circuit underlying a characteristic rolling response that larvae exhibit when evading parasitization by wasps, potentially regulates the termination of rolling and the subsequent transition to escape crawling. We showed that these descending neurons, which we designated as SeIN128, were identical to those previously identified as a component of the nociceptive circuit; were inhibitory neurons that receive excitatory inputs from Basin-2, a second-order interneuron upstream, and A00c, an ascending neuron downstream of Basin-2; and provided GABAergic feedback onto Basin-2, presumably via the axo-axonal synaptic contacts made by the axon terminal endings of SeIN128 neurons onto the axons of Basin-2. Optogenetic activation studies further showed that co-activation of SeIN128 and Basin-2 neurons systematically altered the temporal dynamics of rolling and subsequent escape crawling. Collectively, the evidence suggests that the ensemble of SeIN128, Basin-2, and A00c neurons constitutes a novel inhibitory feedback circuit that reduces Basin-2 activity, which in turn, influences the activity of a key interneuron of the rolling circuit via a novel inhibitory mechanism.

### Feedback inhibition in a nociceptive circuit

Feedback inhibition occurs when an excitatory neuron sends projections to an inhibitory neuron, which in turn sends projections back onto the same excitatory neuron, often at its presynaptic terminals (*Isaacson and Scanziani, 2011*; *Kapfer et al., 2007*; *Ray et al., 2020*; *Stokes and Isaacson, 2010*; *Yoshimura and Callaway, 2005*). The hallmark of feedback inhibition lies in its ability to modulate the duration and magnitude of incoming excitatory signals, thereby fine-tuning neural responses and maintaining homeostasis (*Kapfer et al., 2007*; *Papadopoulou et al., 2011*; *Stokes and Isaacson, 2010*; *Yoshimura and Callaway, 2005*). Compared to the fast temporal dynamics of feedforward inhibition, in which an inhibitory neuron directly inhibits an excitatory neuron downstream of it, the

temporal dynamics of feedback inhibition are slower, primarily due to the added synaptic delays (two or more) following activation of an excitatory neuron (*Papadopoulou et al., 2011*; *Ray et al., 2020*; *Stokes and Isaacson, 2010*). The slow temporal dynamics serve to inhibit the sustained neural activity and magnitude of incoming excitatory signals (*Papadopoulou et al., 2011*; *Ray et al., 2020*; *Stokes and Isaacson, 2010*).

In this study, we showed that SeIN128 neurons are descending neurons whose main inputs arrive in the brain and SEZ regions, and whose outputs target the VNC. We also found that SeIN128 neurons receive excitatory inputs from Basin-2 as well as its downstream neuron A00c, and in turn send inhibitory projections back to these neurons in the VNC, potentially establishing a feedback inhibition motif that modulates the nociceptive rolling circuit. The interplay we observed among SeIN128 neurons, Basin-2, and A00c is consistent with this view. Our findings revealed that activation of SeIN128 neurons has a suppressive effect on Basin-2 activity and, notably, on the duration of rolling. These observations support the idea that feedback inhibition is critical in regulating the temporal aspects of nociceptive responses.

## Inhibition of Basin-2 by SeIN128 neurons is mediated by axo-axonal synapses

Neurons form a wide variety of neural networks that perform various computations in the brain. Typically, a neuron receives inputs via axo-dendritic synapses (i.e., contacts made by the axon terminals of an upstream neuron with its dendrites), which play a role in the spatial and temporal computations that lead to the firing of action potentials. Less commonly, the axon terminals of an upstream neuron may contact the soma (i.e., via axo-somatic synapses) or axon (i.e., via axo-axonal synapses) of a downstream neuron (*Palay, 1956*; *Pinault et al., 1997*; *Zheng et al., 2018*). Axo-axonal synapses have a subtle effect on neurotransmission at the network level because the activity in presynaptic neurons does not alter the membrane potential (*Cattaert and El Manira, 1999*; *Guo and Hu, 2014*; *McGann, 2013*). Axo-axonal synapses mainly affect the release probability of neurotransmitter vesicles in response to an action potential triggered in the postsynaptic neuron (*McGann, 2013*; *Oleson et al., 2012*).

Recent studies suggest that the activity of axo-axonal synapses can prevent the transmission of action potentials. For example, it has been reported that, neurotransmission mediated by type-B muscarinic receptors at lateral axo-axonal connections between *Drosophila* Kenyon cells is critical for stimulus specificity learning in *Drosophila Manoim et al., 2022*; inhibitory axo-axonal connections between Chandelier cells and CA1 pyramidal cells are important for activity-dependent plasticity (*Pan-Vazquez et al., 2020*; *Schneider-Mizell et al., 2021*); and GABAergic axo-axonal interneurons in the amygdala are crucial for generating action potentials in the principal output cells (*Veres et al., 2023*). Furthermore, EM connectome analyses of the entire larval brain reveal that ~70% of all synapses in *Drosophila* larvae are axo-dendritic whereas ~30% are axo-axonal, suggesting that the latter may have considerable influence over network function (*Winding et al., 2023*).

In this study, we found a feedback connection between SeIN128 and Basin-2 mediated by axon-axonal synapses (*Figure 4—figure supplement 2E–G*). The slow increase of SeIN128 activity in response to Basin-2 or A00c activation could potentially occur because of these axo-axonal connections. Alternatively, the slow response in SeIN128 may involve as yet unidentified indirect connections from Basin-2 or A00c to the main inputs in the SEZ region. This delayed activity may play an important role in the feedback inhibition of Basin-2 activity and consequently in the termination of rolling.

## Role of SeIN128 in other escape behaviors

Although the current study focused on rolling, activation of SS04185 neurons appeared to influence other escape behaviors. First, *during* co-activation of SS04185 and Basins (*Figure 1B*, lower panel), the frequency of crawling following the initial rolling bout increased, whereas when only the Basins were activated, animals typically showed multiple rolling bouts during the 30-s stimulation period (*Figure 1B*, upper panel). This observation might be attributed to the strong stimulation induced by Chrimson activation of Basins alone that interrupts crawling via the intrusion of repeated rolling bouts, compared with the co-activation of SS04185 neurons and Basins that reduces the intrusion of rolling. More notably, *after* either co-activation of SS04185 neurons and Basins (*Figure 1B*, lower panel) or activation of SS04185 neurons alone (data not shown), the frequency of turning increased *upon the*

*cessation (offset) of stimulation*, but not following the activation of Basins alone (*Figure 1B*, upper panel). This second observation suggests the possibility that activation of SS04185 neurons leads to sustained inhibition of turning throughout the stimulation period, which when released upon the offset of stimulation, results in a rebound in the frequency of turning beyond baseline levels. Alternatively, activation of SS04185-MB neurons alone may independently trigger the increase in turning frequency following the offset of stimulation (*Figure 1B*, *Figure 2—figure supplement 1K*). A comprehensive examination of this question, however, is beyond the scope of the present study.

## Roles of Basin-2 and Basin-4 in escape behavior

Previous studies have shown that, Basin-2 and Basin-4 receive both chordotonal sensory and nociceptive sensory inputs, and in addition, play a critical role in escape behavior (*Ohyama et al., 2015*). Here, we investigated the differences between rolling induced by activation of Basin-2 or Basin-4. We found that activation of Basin-2 induced rolling that was sustained. Furthermore, activation of SeIN128 neurons reduced the duration of rolling induced by co-activation of Basin-2, which resulted in a delay in the onset of rolling and an earlier termination of rolling. These data indicate that activation of Basin-2 serves to maintain rolling. Connectome data indicate that SeIN128 neurons provide inhibitory input onto Basin-2, which is consistent with the finding that SeIN128 activation reduces the duration of rolling.

On the other hand, activation of Basin-4 induced rolling that was transient, which was then followed by rapid crawling. Furthermore, activation of SeIN128 neurons reduced the probability of rolling but did not affect the duration of rolling (*Figure 8F*). This suggests that activation of Basin-4 is important for the induction of rolling, but not its maintenance. The behavioral effects of co-activating SeIN128 and Basin-4, together with connectome data indicating the lack of connections between SeIN128 neurons and Basin-4, suggest that these descending neurons target neurons downstream of Basin-4 neurons.

## Other inputs onto SeIN128 neurons modify escape behavior

The dendritic regions of SeIN128 neurons are located in the SEZ and brain, suggesting that SeIN128 neurons receive other inputs from the SEZ and brain neurons. In this study, we did not examine these inputs. Connectome data indicate that MB output neurons project onto SeIN128 neurons (*Ohyama et al., 2015*). Given the well-established role of MB neurons in learning, this finding suggests that SeIN128 neurons could play a role in experience-dependent modulation of rolling. Two recent studies have shown that descending neurons inhibit nociceptive neurons (*Nakamizo-Dojo et al., 2023*; *Oikawa et al., 2023*). Specifically, one study showed that insulin signaling modulates escape behavior by activating GABAergic descending neurons that inhibit nociceptive sensory neurons (*Nakamizo-Dojo et al., 2023*), whereas the other demonstrated an inhibitory mechanism mediated by the neuropeptide Drosulfakinin, a homolog of cholecystokinin in mammals (*Oikawa et al., 2023*). Whether SeIN128 neurons are also influenced by insulin signaling or Drosulfakinin, however, remains to be seen.

## Implications for mechanistic analyses of behavioral sequences

In this study, we did not investigate how crawling is initiated after rolling. Recent studies of the motor circuits underlying rolling and crawling suggest that different premotor neurons are involved in driving each action (*Cooney et al., 2023*; *Kohsaka, 2023*; *Zarin et al., 2019*). When we co-activated SS04185 and Basins, the duration of rolling decreased and the latency to onset of crawling decreased. These data are consistent with the notion that the rolling circuit inhibits the crawling circuit. It would be of interest to examine if premotor crawling neurons are inhibited during rolling, and if so, how they are activated following Basin inhibition to trigger crawling.

In summary, our study delineates a neuronal ensemble consisting of a set of descending inhibitory neurons, a first-order interneuron (Basin-2), and an ascending neuron (A00c) in fruit fly larvae, which functions as an inhibitory feedback circuit that regulates the probability and duration of rolling, and thereby facilitates the transition from rolling to crawling. This work represents another example of how detailed analyses of connectomes and functional analyses of neural and behavioral activity can identify mechanistic explanations of behavioral phenomena at the level of neural circuits—in this case, how neuronal ensembles generate behavioral sequences.

# Materials and methods

## Key resources table

| Reagent type (species) or resource | Designation | Source or reference | Identifiers | Additional information |
|---|---|---|---|---|
| Gene (*Drosophila melanogaster*) | Killer Zipper (KZip⁺) | *Dolan et al., 2017* Bloomington *Drosophila* Stock Center (BDSC) | RRID:BDRC_76254 | |
| Genetic reagent (*D. melanogaster*) | R72F11-Gal4 | BDSC | BDSC:39786; RRID:BDRC_39786 | FlyBase: P{GMR72F11-GAL4} |
| Genetic reagent (*D. melanogaster*) | R71A10-Gal4 | BDSC | BDSC:39562; RRID:BDRC_39562 | FlyBase P{GMR71A10-GAL4} |
| Genetic reagent (*D. melanogaster*) | SS04185-Gal4 | Gift from Zlatic laboratory | N/A | R54B01-Gal4$^{AD}$; R46E07-Gal4$^{DBD}$ |
| Genetic reagent (*D. melanogaster*) | SS00739-Gal4 | Gift from Zlatic laboratory | N/A | R72F11-Gal4$^{AD}$;R38H09-Gal4$^{DBD}$ |
| Genetic reagent (*D. melanogaster*) | SS00740-Gal4 | Gift from Zlatic laboratory | N/A | R72F11-Gal4$^{AD}$;R57F07-Gal4$^{DBD}$ |
| Genetic reagent (*D. melanogaster*) | MB247-Gal4 | BDSC | BDRC:50742; RRID:BDRC_50742 | FlyBase P{Mef2-GAL4.247} |
| Genetic reagent (*D. melanogaster*) | R13F02-LexA | BDSC | BDRC:52460; RRID:BDRC_52460 | FlyBase P{GMR13F02-lexA} |
| Genetic reagent (*D. melanogaster*) | R72F11-LexA | BDSC | BDRC:94661; RRID:BDRC_94661 | FlyBase P{GMR72F11-lexA} |
| Genetic reagent (*D. melanogaster*) | R71A10-LexA | Gift from Zlatic lab | N/A | |
| Genetic reagent (*D. melanogaster*) | Mi{Trojan-LexA-QFAD.2}Gad1 | BDSC | BDRC:60324; RRID:BDRC_60324 | FlyBase Mi{Trojan-lexA:QFAD.2} |
| Genetic reagent (*D. melanogaster*) | 20xUAS-IVS-CsChrimson::mVenus | BDSC | BDRC:55134; RRID:BDRC_55134 | FlyBase P{20XUAS-IVS-CsChrimson.mVenus} |
| Genetic reagent (*D. melanogaster*) | 20xUAS-IVS-CsChrimson::mVenus | BDSC | BDRC:55136; RRID:BDRC_55136 | FlyBase P{20XUAS-IVS-CsChrimson.mVenus} |
| Genetic reagent (*D. melanogaster*) | 13xLexAop2-IVS-CsChrimson::mVenus | BDSC | BDRC:55137; RRID:BDRC_55137 | FlyBase P{13XLexAop2-IVS-CsChrimson.mVenus} |
| Genetic reagent (*D. melanogaster*) | 13xLexAop2-IVS-CsChrimson::tdTomato | Gift from Rubin lab | N/A | |
| Genetic reagent (*D. melanogaster*) | 13xLexAop2-IVS-CsChrimson::tdTomato | BDSC | BDRC:82183; RRID:BDRC_82183 | FlyBase PBac{13XLexAop2-IVS-CsChrimson.tdTomato} |
| Genetic reagent (*D. melanogaster*) | 20xUAS(FRT.stop)CsChrimson.mVenus(attP18), pBPhsFlp2::Pest | Gift from Rubin lab | N/A | |
| Genetic reagent (*D. melanogaster*) | hs(KDRT.stop)FLP | BDSC | BDRC:67091; RRID:BDRC_67091 | FlyBase symbol: P{hs(KDRT.stop)FLP} |
| Genetic reagent (*D. melanogaster*) | 20xUAS(FRT.stop)CsChrimson::mVenus | Gift from Rubin lab | N/A | |
| Genetic reagent (*D. melanogaster*) | UAS-TeTxLC.tnt | BDSC | BDRC:28838; RRID:BDRC_28838 | FlyBase symbol: P{UAS-TeTxLC.tnt} |
| Genetic reagent (*D. melanogaster*) | 20xUAS-TTS-Shibire$^{ts1}$-p10 | BDSC | BDRC:66600; RRID:BDRC_66600 | FlyBase PBac{20XUAS-TTS-shi$^{ts1}$-p10} |
| Genetic reagent (*D. melanogaster*) | 10xUAS-IVS-mry::GFP | Gift from Rubin lab | N/A | |
| Genetic reagent (*D. melanogaster*) | 13xLexAop-dsRed | Gift from Rubin lab | N/A | |
| Genetic reagent (*D. melanogaster*) | 20xUAS-IVS-GCaMP6s | BDSC | BDRC:42749; RRID:BDRC_42749 | FlyBase PBac{20XUAS-IVS-GCaMP6s} |
| Genetic reagent (*D. melanogaster*) | 20xLexAop-IVS-Syn21-GCaMP6s | Gift from Rubin lab | N/A | |

*Continued on next page*

*Continued*

| Reagent type (species) or resource | Designation | Source or reference | Identifiers | Additional information |
|---|---|---|---|---|
| Genetic reagent (*D. melanogaster*) | 20xUAS-Syn21-opGCaMP6s | Gift from Rubin lab | N/A | |
| Genetic reagent (*D. melanogaster*) | 10xUAS-Syn21-CsChrimson88::tdTomato | Gift from Rubin lab | N/A | |
| Genetic reagent (*D. melanogaster*) | HMS02355 | BDSC | BDRC:41958; RRID:BDRC_41958 | FlyBase P{TRiP.HMS02355} |
| Genetic reagent (*D. melanogaster*) | HMC03388 | BDSC | BDRC:51817; RRID:BDRC_51817 | FlyBase P{TRiP.HMC03388} |
| Genetic reagent (*D. melanogaster*) | JF02989 | BDSC | BDRC:28353; RRID:BDRC_28353 | FlyBase P{TRiP.JF02989} |
| Genetic reagent (*D. melanogaster*) | HMC02975 | BDSC | BDRC:50608; RRID:BDRC_50608 | FlyBase P{TRiP.HMC02975} |
| Genetic reagent (*D. melanogaster*) | HMC03643 | BDSC | BDRC:52903; RRID:BDRC_52903 | FlyBase P{TRiP.HMC03643} |
| Antibody | anti-Brp, clone nc82 (Mouse monoclonal) | Developmental Studies Hybridoma Bank | Cat#: nc82, RRID:AB_2314866 | IHC (1:50) |
| Antibody | 1D4 anti-fasciclin II (Mouse monoclonal) | Developmental Studies Hybridoma Bank | Cat#: 1D4 anti-Fasciclin II, RRID:AB_528235 | IHC (1:20) |
| Antibody | anti-cadherin, DN-(extracellular domain) (Rat monoclonal) | Developmental Studies Hybridoma Bank | Cat#: DN-Ex #8, RRID:AB_528121 | IHC (1:20) |
| Antibody | anti-GFP (Chicken polyclonal) | Abcam | Cat#:ab13970, RRID:AB_300798 | IHC (1:3000) |
| Antibody | anti-GFP (Rabbit polyclonal) | Thermo Fisher Scientific | Cat#:A-6455, RRID:AB_221570 | IHC (1:1000) |
| Antibody | anti-DsRed (Rabbit polyclonal) | Takara Bio | Cat#:632496, RRID:AB_10013483 | IHC (1:1000) |
| Antibody | anti-*Drosophila* choline acetyltransferase (Mouse monoclonal) | Developmental Studies Hybridoma Bank | Cat#:chat4b1, RRID:AB_528122 | IHC (1:20) |
| Antibody | anti-GABA (Rabbit polyclonal) | Millipore Sigma | Cat#:A2052 | IHC (1:1000) |
| Antibody | *Drosophila* VGLUT (Rabbit polyclonal) | Gift from McCabe laboratory; *Banerjee et al., 2021* | | IHC (1:20) |
| Antibody | anti-VGAT (Rabbit polyclonal) | Gift from Krantz laboratory; *Fei et al., 2010* | | IHC (1:200) |
| Antibody | anti-Elav (Rat polyclonal) | Developmental Studies Hybridoma Bank | Cat#:7E8A10, RRID:AB_528218 | IHC (1:50) |
| Antibody (secondary) | anti-chicken IgY (H+L) Alexa Fluor 488 (Goat polyclonal) | Thermo Fisher Scientific | Cat#:A-11039, RRID:AB_2534096 | IHC (1:500) |
| Antibody (secondary) | anti-rabbit IgG (H+L) Alexa Fluor 488 (Goat polyclonal) | Thermo Fisher Scientific | Cat#:A-11034, RRID:AB_2576217 | IHC (1:500) |
| Antibody (secondary) | anti-rabbit IgG (H+L) Alexa Fluor 568 (Goat polyclonal) | Thermo Fisher Scientific | Cat#:A-11011, RRID:AB_143157 | IHC (1:500) |
| Antibody (secondary) | anti-mouse IgG (H+L) Alexa Fluor 568 (Goat polyclonal) | Thermo Fisher Scientific | Cat#:A-11004, RRID:AB_2534072 | IHC (1:500) |
| Antibody (secondary) | anti-rat IgG (H+L) Alexa Fluor 568 (Goat polyclonal) | Thermo Fisher Scientific | Cat#:A-11077, RRID:AB_2534121 | IHC (1:500) |
| Chemical compound, drug | PBS, phosphate-buffered saline, 10× solution | Fisher Scientific | Cat#:BP399-1 | |
| Chemical compound, drug | Triton X-100 | Millipore Sigma | Cat#:X100-100ML | |

*Continued on next page*

*Continued*

| Reagent type (species) or resource | Designation | Source or reference | Identifiers | Additional information |
|---|---|---|---|---|
| Chemical compound, drug | Paraformaldehyde 20% aqueous solution | Electron Microscopy Sciences | Cat#:15713 | |
| Chemical compound, drug | Normal goat serum | Gibco | Cat#:PCN5000 | |
| Chemical compound, drug | VECTASHIELD antifade mounting medium | Vector Laboratories | Cat#:H-1000-10 | |
| Chemical compound, drug | *Drosophila* Agar | Diamed | Cat#:GEN66-103 | |
| Chemical compound, drug | All Trans Retinal | Toronto Research Chemicals Inc | Cat#:R24000 | |
| Chemical compound, drug | Poly-L-lysine | Sigma-Aldrich | Cat#:P1524 | |
| Software, algorithm | FIJI | https://fiji.sc/ | RRID:SCR_002285 | |
| Software, algorithm | MATLAB | MathWorks | RRID:SCR_001622 | |
| Software, algorithm | CATMAID | https://catmaid.readthedocs.org/ | RRID:SCR_006278 | |
| Software, algorithm | Multi Worm Tracker | http://sourceforge.net/projects/mwt | N/A | |
| Software, algorithm | ZEN | Carl Zeiss Microscopy | Version 2.1 | |
| Software, algorithm | Affinity Designer | Affinity | Version 1.10.5 | |
| Software, algorithm | ScanImage | MBF Bioscience | N/A | |

## Fly stocks and maintenance

All *D. melanogaster* stock lines used in this study were raised on Bloomington *Drosophila* Stock Center cornmeal food. Flies were maintained in a humidity- and temperature-controlled chamber kept at 18 or 25°C, 40% humidity, and set to a 12-hr light/dark cycle. All crosses for experiments were reared at 25°C and 40% humidity.

## Heat shock FlpOut mosaic expression

First instar *Drosophila* larvae were heat shocked in water bath at 37°C for 12 min as previously reported (*Nern et al., 2015*). With the precise temporal and temperature control of heat shock, larvae with the genotype of *w⁺, hs(KDRT.stop)FLP/13xLexAop2-IVS-CsChrimson::tdTomato; R54B01-Gal4.AD/72F11-LexA;20xUAS-(FRT.stop)-CsChrimson::mVenus/R46E07-Gal4.DBD* had sporadic CsChrimson::mVenus expression driven by SS04185 split GAL4. As a result, the ratio of the larvae with SS04185-DN and SS04185-MB expression to those with only SS04185-MB expression was 1:1. Each individual larva was individually examined with optogenetic stimulation and behavior analysis. After behavioral experiments, mVenus expression in CNS was confirmed under the fluorescence microscope.

## Fly genotypes used in experiments
### Main figures

| Fig. | Panel | Labels | Genotypes |
|---|---|---|---|
| 1 | B | Basins>Chrimson | *20xUAS-IVS-CsChrimson::mVenus/+;+; R72F11-Gal4/+* |
| 1 | B | Basins+SS04185>Chrimson | *20xUAS-IVS-CsChrimson::mVenus/+; R54B01-Gal4.AD/+; R46E07-Gal4.DBD/ R72F11-Gal4* |
| 1 | C, D, F | control | *20xUAS-IVS-CsChrimson::mVenus/+; +; R72F11-Gal4/+* |
| 1 | C, D, F | SS04185 | *20xUAS-IVS-CsChrimson::mVenus/+; R54B01-Gal4.AD/+; R46E07-Gal4.DBD/R72F11-Gal4* |
| 1 | E | ctrl/attp2>Chrimson | *20xUAS-IVS-CsChrimson::mVenus/+;;* |

*Continued on next page*

*Continued*

| Fig. | Panel | Labels | Genotypes |
|---|---|---|---|
| 1 | E | SS04185/attp2>Chrimson | *20xUAS-IVS-CsChrimson::mVenus/+; R54B01-Gal4.AD/+; R46E07-Gal4.DBD/+* |
| 1 | E | ctrl/Basins>Chrimson | *20xUAS-IVS-CsChrimson::mVenus/+; +; R72F11-Gal4/+* |
| 1 | E | SS04185/Basins>Chrimson | *20xUAS-IVS-CsChrimson::mVenus/+; R54B01-Gal4.AD/+; R46E07-Gal4.DBD/R72F11-Gal4* |
| 1 | G–I | ctrl | *20xUAS-IVS-CsChrimson::mVenus/+; +; R72F11-Gal4/+* |
| 1 | G–I | SS04185 | *20xUAS-IVS-CsChrimson::mVenus/+; R54B01-Gal4.AD/+; R46E07-Gal4.DBD/R72F11-Gal4* |
| 2 | A | | *10xUAS-IVS-myr::GFP/+; R54B01-Gal4.AD/+; R46E07-Gal4.DBD/+* |
| 2 | B | control | *20xUAS-IVS-CsChrimson::mVenus/+; R54B01-Gal4.AD/+; R46E07-Gal4.DBD/R72F11-Gal4* |
| 2 | B | MB>KZip+ | *20xUAS-IVS-CsChrimson::mVenus/+; R13F02-LexA,LexAop-KZip⁺/ R54B01-Gal4.AD; R72F11-Gal4/R46E07-Gal4.DBD* |
| 2 | C, E | MB>KZip+/ctrl | *20xUAS-IVS-CsChrimson::mVenus/+; R13F02-LexA,LexAop-KZip⁺/+; R72F11-Gal4/+* |
| 2 | C, E | -/SS04185 | *20xUAS-IVS-CsChrimson::mVenus/+; R54B01-Gal4.AD/+; R46E07-Gal4.DBD/R72F11-Gal4* |
| 2 | C, E | MB>KZip+/SS04185 | *20xUAS-IVS-CsChrimson::mVenus/+; R13F02-LexA,LexAop-KZip⁺/ R54B01-Gal4.AD; R72F11-Gal4/R46E07-Gal4.DBD* |
| 2 | D | MB>KZip+ | *20xUAS-IVS-CsChrimson::mVenus/+; R13F02-LexA,LexAop-KZip⁺/+; R72F11-Gal4/+* |
| 2 | D | SS04185 | *20xUAS-IVS-CsChrimson::mVenus/+; R54B01-Gal4.AD/+; R46E07-Gal4.DBD/R72F11-Gal4* |
| 2 | D | MB>KZip+, SS04185 | *20xUAS-IVS-CsChrimson::mVenus/+; R13F02-LexA,LexAop-KZip⁺/ R54B01-Gal4.AD; R72F11-Gal4/R46E07-Gal4.DBD* |
| 2 | F, H | ctrl | *w+, hs(KDRT.stop)FLP/13xLexAop2-IVS-CsChrimson::tdTomato; R54B01-Gal4.AD/72F11-LexA;20xUAS-(FRT.stop)-CsChrimson::mVenus/ R46E07-Gal4.DBD* |
| 2 | F, H | SS04185-DN | *w+, hs(KDRT.stop)FLP/13xLexAop2-IVS-CsChrimson::tdTomato; R54B01-Gal4.AD/72F11-LexA;20xUAS-(FRT.stop)-CsChrimson::mVenus/ R46E07-Gal4.DBD* |
| 2 | G | control | *w+, hs(KDRT.stop)FLP/13xLexAop2-IVS-CsChrimson::tdTomato; R54B01-Gal4.AD/72F11-LexA;20xUAS-(FRT.stop)-CsChrimson::mVenus/ R46E07-Gal4.DBD* |
| 2 | G | SS04185-DN | *w+, hs(KDRT.stop)FLP/13xLexAop2-IVS-CsChrimson::tdTomato; R54B01-Gal4.AD/72F11-LexA;20xUAS-(FRT.stop)-CsChrimson::mVenus/ R46E07-Gal4.DBD* |
| 3 | D, E | | *10xUAS-IVS-myr::GFP/+; R54B01-Gal4.AD/+; R46E07-Gal4.DBD/+* |
| 3 | F | | *w; R54B01-Gal4.AD/R72F11-LexA; R46E07-Gal4.DBD/13xLexAop2-IVS-CsChrimson::tdTomato, 20xUAS-IVS-GCaMP6s* |
| 3 | H | | *w; R54B01-Gal4.AD/R71A10-LexA; R46E07-Gal4.DBD/13xLexAop2-IVS-CsChrimson::tdTomato, 20xUAS-IVS-GCaMP6s* |
| 3 | J | | *w; R54B01-Gal4.AD/ppk1.9-LexA; R46E07-Gal4.DBD/13xLexAop2-IVS-CsChrimson::tdTomato, 20xUAS-IVS-GCaMP6s* |
| 4 | B | | *w; R72F11-LexA/R54B01-Gal4.AD; 13xLexAop-CsChrimson, 20xUAS-IVS-UAS-GCaMP6s/R46E07-Gal4.DBD* |
| 4 | C | | *w; R71A10-LexA/R54B01-Gal4.AD; 13xLexAop-CsChrimson, 20xUAS-IVS-UAS-GCaMP6s/R46E07-Gal4.DBD* |

*Continued on next page*

*Continued*

| Fig. | Panel | Labels | Genotypes |
|------|-------|--------|-----------|
| 4 | D | | *w; R72F11-LexA/+; 13xLexAop2-IVS -CsChrimson::tdTomato, 20xUAS-IVS-UAS-GCaMP6s/R71A10-Gal4* |
| 4 | E | A00c | *w; R72F11-LexA/+; 13xLexAop2-IVS -CsChrimson::tdTomato, 20xUAS-IVS-UAS-GCaMP6s/R71A10-Gal4* |
| 4 | E | SS04185 | *w; R72F11-LexA/R54B01-Gal4.AD; 13xLexAop2-IVS -CsChrimson::tdTomato, 20xUAS-IVS-UAS-GCaMP6s/R46E07-Gal4.DBD* |
| 5 | A | | *10xUAS-myr::GFP; R54B01-Gal4.AD/13x-LexAop-dsRed; R46E07-Gal4.DBD/ Mi{Trojan-LexA-QFAD.2}Gad1* |
| 5 | B, D | control | *13xLexAop2-IVS-CsChrimson::mVenus;R72F11-lexA/+; HMS02355/+* |
| 5 | B, D | SS04185 | *13xLexAop2-IVS-CsChrimson::mVenus; R72F11-lexA/R54B01-Gal4.AD; HMS02355/R46E07-Gal4.DBD* |
| 5 | C | ctrl | *13xLexAop2-IVS-CsChrimson::mVenus;R72F11-lexA/+; HMS02355/+* |
| 5 | C | SS04185 | *13xLexAop2-IVS-CsChrimson::mVenus; R72F11-lexA/R54B01-Gal4.AD; HMS02355/R46E07-Gal4.DBD* |
| 6 | A | control>TNT | *13xLexAop2-IVS-CsChrimson::mVenus; R72F11-LexA/+; UAS-TeTxLC.tnt /+* |
| 6 | A | SS04185>TNT | *13xLexAop2-IVS-CsChrimson::mVenus; R72F11-LexA/R54B01-Gal4.AD; UAS-TeTxLC.tnt/R46E07-Gal4.DBD* |
| 6 | B, D–F | ctrl | *13xLexAop2-IVS-CsChrimson::mVenus; R72F11-LexA/+; UAS-TeTxLC.tnt /+* |
| 6 | B, D–F | SS04185 | *13xLexAop2-IVS-CsChrimson::mVenus; R72F11-LexA/R54B01-Gal4.AD; UAS-TeTxLC.tnt/R46E07-Gal4.DBD* |
| 6 | C | control | *13xLexAop2-IVS-CsChrimson::mVenus; R72F11-LexA/+; UAS-TeTxLC.tnt /+* |
| 6 | C | SS04185 | *13xLexAop2-IVS-CsChrimson::mVenus; R72F11-LexA/R54B01-Gal4.AD; UAS-TeTxLC.tnt/R46E07-Gal4.DBD* |
| 7 | A, B | control | *20xUAS-IVS-CsChrimson::mVenus/+;; R72F11-Gal4/+* |
| 7 | A, B | GABA-B-R1[1] | *20xUAS-IVS-CsChrimson::mVenus/+;; R72F11-Gal4/UAS-HMC03388* |
| 7 | A, B | GABA-B-R1[2] | *20xUAS-IVS-CsChrimson::mVenus/+;; R72F11-Gal4/UAS-JF02989* |
| 7 | A, B | GABA-B-R2 | *20xUAS-IVS-CsChrimson::mVenus/+;; R72F11-Gal4/UAS-HMC02975* |
| 7 | A, B | GABA-A-R | *20xUAS-IVS-CsChrimson::mVenus/+; R72F11-Gal4/UAS-HMC03643* |
| 7 | C | control | *20xUAS-Syn21-opGCaMP6s,10xUAS-Syn21-CsChrimson88::tdTomato/+;CyO/+;TM6/R72F11-Gal4* |
| 7 | C | SS04185 | *20xUAS-Syn21-opGCaMP6s,10xUAS-Syn21-CsChrimson88::tdTomato/+;CyO/R54B01-Gal4.AD;R72F11-Gal4/ R46E07-Gal4.DBD* |
| 8 | A, C, D | ctrl | *20xUAS-IVS-CsChrimson::mVenus/+; R72F11-Gal4.AD/+; R38H09-Gal4.DBD/+* |
| 8 | A, C, D | SS04185 | *20xUAS-IVS-CsChrimson::mVenus/+; R72F11-Gal4.AD/R54B01-Gal4.AD; R38H09-Gal4.DBD/R46E07-Gal4.DBD* |
| 8 | B | control | *20xUAS-IVS-CsChrimson::mVenus/+; R72F11-Gal4.AD/+; R38H09-Gal4.DBD/+* |
| 8 | B | SS04185 | *20xUAS-IVS-CsChrimson::mVenus/+; R72F11-Gal4.AD/R54B01-Gal4.AD; R38H09-Gal4.DBD/R46E07-Gal4.DBD* |
| 8 | E, G, H | ctrl | *20xUAS-IVS-CsChrimson::mVenus/+; R72F11-Gal4.AD/+; R57F07-Gal4.DBD/+* |

*Continued on next page*

*Continued*

| Fig. | Panel | Labels | Genotypes |
|---|---|---|---|
| 8 | E, G, H | SS04185 | *20xUAS-IVS-CsChrimson::mVenus/+; R72F11-Gal4.AD/R54B01-Gal4.AD; R57F07-Gal4.DBD/R46E07-Gal4.DBD* |
| 8 | F | control | *20xUAS-IVS-CsChrimson::mVenus/+; R72F11-Gal4.AD/+; R57F07-Gal4.DBD/+* |
| 8 | F | SS04185 | *20xUAS-IVS-CsChrimson::mVenus/+; R72F11-Gal4.AD/R54B01-Gal4.AD; R57F07-Gal4.DBD/R46E07-Gal4.DBD* |

## Figure supplements

| Fig. | Panel | Labels | Genotypes |
|---|---|---|---|
| 1–1 | A–D | ctrl | *20xUAS-IVS-CsChrimson::mVenus/+;;* |
| 1–1 | A–D | SS04185 | *20xUAS-IVS-CsChrimson::mVenus/+; R54B01-Gal4.AD/+; R46E07-Gal4.DBD/+* |
| 1–1 | E, G, H | ctrl | *20xUAS-IVS-CsChrimson::mVenus/+; +; R72F11-Gal4/+* |
| 1–1 | E, G, H | SS04185 | *20xUAS-IVS-CsChrimson::mVenus/+; R54B01-Gal4.AD/+; R46E07-Gal4.DBD/R72F11-Gal4* |
| 1–1 | F | control | *20xUAS-IVS-CsChrimson::mVenus/+; +; R72F11-Gal4/+* |
| 1–1 | F | SS04185 | *20xUAS-IVS-CsChrimson::mVenus/+; R54B01-Gal4.AD/+; R46E07-Gal4.DBD/R72F11-Gal4* |
| 1–1 | I | ctrl/attp2>Chrimson | *20xUAS-IVS-CsChrimson::mVenus/+;;* |
| 1–1 | I | SS04185/attp2>Chrimson | *20xUAS-IVS-CsChrimson::mVenus/+; R54B01-Gal4.AD/+; R46E07-Gal4.DBD/+* |
| 1–1 | I | ctrl/Basins>Chrimson | *20xUAS-IVS-CsChrimson::mVenus/+; +; R72F11-Gal4/+* |
| 1–1 | I | SS04185/Basins>Chrimson | *20xUAS-IVS-CsChrimson::mVenus/+; R54B01-Gal4.AD/+; R46E07-Gal4.DBD/R72F11-Gal4* |
| 1–2 | A–C | control | *20xUAS-IVS-CsChrimson::mVenus/+;; R72F11-Gal4/+* |
| 1–2 | A–C | 54B01-AD | *20xUAS-IVS-CsChrimson::mVenus/+; R54B01-Gal4.AD/+; R72F11-Gal4/+* |
| 1–2 | A–C | 46E07-DBD | *20xUAS-IVS-CsChrimson::mVenus/+; +; R72F11-Gal4/R46E07-Gal4.DBD* |
| 1–2 | A–C | SS04185 | *20xUAS-IVS-CsChrimson::mVenus/+; R54B01-Gal4.AD/+; R46E07-Gal4.DBD/R72F11-Gal4* |
| 2 | A | | *20xUAS-IVS-CsChrimson::mVenus/+; R13F02-LexA,LexAop-KZip⁺/R54B01-Gal4.AD; R72F11-Gal4/R46E07-Gal4.DBD* |
| 2 | B | MB>Kzip+ | *20xUAS-IVS-CsChrimson::mVenus/+; R13F02-LexA,LexAop-Kzip⁺/+; R72F11-Gal4/+* |
| 2 | B | SS04185 | *20xUAS-IVS-CsChrimson::mVenus/+; R54B01-Gal4.AD/+; R46E07-Gal4.DBD/R72F11-Gal4* |
| 2 | B | MB>Kzip+, SS04185 | *20xUAS-IVS-CsChrimson::mVenus/+; R13F02-LexA,LexAop-Kzip⁺/R54B01-Gal4.AD; R72F11-Gal4/R46E07-Gal4.DBD* |
| 2 | C | MB>Kzip+/ctrl | *20xUAS-IVS-CsChrimson::mVenus/+; R13F02-LexA,LexAop-Kzip⁺/+; R72F11-Gal4/+* |
| 2 | C | -/SS04185 | *20xUAS-IVS-CsChrimson::mVenus/+; R54B01-Gal4.AD/+; R46E07-Gal4.DBD/R72F11-Gal4* |
| 2 | C | MB>Kzip+/SS04185 | *20xUAS-IVS-CsChrimson::mVenus/+; R13F02-LexA,LexAop-Kzip⁺/R54B01-Gal4.AD; R72F11-Gal4/R46E07-Gal4.DBD* |

*Continued on next page*

*Continued*

| Fig. | Panel | Labels | Genotypes |
|------|-------|--------|-----------|
| 2 | D | control | *20xUAS-IVS-CsChrimson::mVenus/+; +; R72F11-Gal4/+* |
| 2 | D | MB247 | *20xUAS-IVS-CsChrimson::mVenus/+; R54B01-Gal4.AD/+; R46E07-Gal4.DBD/R72F11-Gal4* |
| 2 | E | ctrl | *20xUAS-IVS-CsChrimson::mVenus/+; +; R72F11-Gal4/+* |
| 2 | E | MB247 | *20xUAS-IVS-CsChrimson::mVenus/+; R54B01-Gal4.AD/+; R46E07-Gal4.DBD/R72F11-Gal4* |
| 2 | F, G | | *w+, hs(KDRT.stop)FLP/13xLexAop-CsChrimson::tdTomato; R54B01-Gal4.AD/72F11-LexA;20xUAS-(FRT.stop)-CsChrimson::mVenus/R46E07-Gal4.DBD* |
| 2 | H | control | *w+, hs(KDRT.stop)FLP/13xLexAop-CsChrimson::tdTomato; R54B01-Gal4.AD/72F11-LexA; 20xUAS-(FRT.stop)-CsChrimson::mVenus/R46E07-Gal4.DBD* |
| 2 | H | SS04185-DN | *w+, hs(KDRT.stop)FLP/13xLexAop-CsChrimson::tdTomato; R54B01-Gal4.AD/72F11-LexA; 20xUAS-(FRT.stop)-CsChrimson::mVenus/R46E07-Gal4.DBD* |
| 2 | I, J | ctrl | *w+, hs(KDRT.stop)FLP/13xLexAop-CsChrimson::tdTomato; R54B01-Gal4.AD/72F11-LexA; 20xUAS-(FRT.stop)-CsChrimson::mVenus/R46E07-Gal4.DBD* |
| 2 | I, J | SS04185-DN | *w+, hs(KDRT.stop)FLP/13xLexAop-CsChrimson::tdTomato; R54B01-Gal4.AD/72F11-LexA; 20xUAS-(FRT.stop)-CsChrimson::mVenus/R46E07-Gal4.DBD* |
| 4 | B | Basins>Chrimson | *w; R72F11-LexA/R54B01-Gal4.AD; 13xLexAop2-IVS-CsChrimson::tdTomato, 20xUAS-IVS- GCaMP6s/R46E07-Gal4.DBD* |
| 4 | B | A00c>Chrimson | *w; R71A10-LexA/R54B01-Gal4.AD; 13xLexAop2-IVS-CsChrimson::tdTomato, 20xUAS-IVS-GCaMP6s/R46E07-Gal4.DBD* |
| 4 | C | SeIN128 (Basins>Chrimson) | *w; R72F11-LexA/R54B01-Gal4.AD; 13xLexAop2-IVS-CsChrimson::tdTomato, 20xUAS-IVS- GCaMP6s/R46E07-Gal4.DBD* |
| 4 | C | SeIN128 (A00c>Chrimson) | *w; R71A10-LexA/R54B01-Gal4.AD; 13xLexAop2-IVS-CsChrimson::tdTomato, 20xUAS-IVS-GCaMP6s/R46E07-Gal4.DBD* |
| 4 | C | A00c (Basins>Chrimson) | *w; R72F11-LexA/+; 13xLexAop2-IVS-CsChrimson::tdTomato, 20xUAS-IVS-GCaMP6s/R71A10-Gal4* |
| 5 | A, B | | *10xUAS-IVS-myr::GFP/+; R54B01-Gal4.AD/+; R46E07-Gal4.DBD/+* |
| 5 | C, D | ctrl | *w;; R57C10-Gal4/+* |
| 5 | C, D | VGAT-RNAi | *w;; R57C10-Gal4/UAS-HMS02355* |
| 6 | A, D | control | *13xLexAop2-IVS-CsChrimson::mVenus; R72F11-LexA/+; 20xUAS-TTS-Shibire[ts1]/+* |
| 6 | A, D | SS04185 | *13xLexAop2-IVS-CsChrimson::mVenus; R72F11-LexA/R54B01-Gal4.AD; 20xUAS-TTS-Shibire[ts1]/R46E07-Gal4.DBD* |
| 6 | B, C, E–G | ctrl | *13xLexAop2-IVS-CsChrimson::mVenus; R72F11-LexA/+; 20xUAS-TTS-Shibire[ts1]/+* |
| 6 | B, C, E–G | SS04185 | *13xLexAop2-IVS-CsChrimson::mVenus; R72F11-LexA/R54B01-Gal4.AD; 20xUAS-TTS-Shibire[ts1]/R46E07-Gal4.DBD* |
| 7 | A–D | control | *20xUAS-IVS-CsChrimson::mVenus/+;; R72F11-Gal4/+* |
| 7 | A–D | GABA-B-R1[1] | *20xUAS-IVS-CsChrimson::mVenus/+;; R72F11-Gal4/UAS-HMC03388* |
| 7 | A–D | GABA-B-R1[2] | *20xUAS-IVS-CsChrimson::mVenus/+;; R72F11-Gal4/UAS-JF02989* |

*Continued*

| Fig. | Panel | Labels | Genotypes |
|---|---|---|---|
| 7 | A–D | GABA-B-R2 | *20xUAS-IVS-CsChrimson::mVenus/+;; R72F11-Gal4/UAS-HMC02975* |
| 7 | A–D | GABA-A-R | *20xUAS-IVS-CsChrimson::mVenus/+;; R72F11-Gal4/UAS-HMC03643* |
| 7 | E | Basins>Chrimson | *20xUAS-Syn21-opGCaMP6s,10xUAS-Syn21-CsChrimson88::tdTomato/+;CyO/+;TM6/R72F11-Gal4* |
| 7 | F | Basins+SeIN128>Chrimson | *20xUAS-Syn21-opGCaMP6s,10xUAS-Syn21-CsChrimson88::tdTomato/+;CyO/R54B01-Gal4.AD;R72F11-Gal4/R46E07-Gal4.DBD* |
| 8 | A | Basin2>Chrimson | *20xUAS-IVS-CsChrimson::mVenus/+; R72F11-Gal4.AD/+; R38H09-Gal4.DBD/+* |
| 8 | B | Basin4>Chrimson | *20xUAS-IVS-CsChrimson::mVenus/+; R72F11-Gal4.AD/+; R57F07-Gal4.DBD/+* |
| 8 | C | Basin-2 | *20xUAS-IVS-CsChrimson::mVenus/+; R72F11-Gal4.AD/+; R38H09-Gal4.DBD/+* |
| 8 | C | Basin-4 | *20xUAS-IVS-CsChrimson::mVenus/+; R72F11-Gal4.AD/+; R57F07-Gal4.DBD/+* |
| 8 | D | control | *20xUAS-IVS-CsChrimson::mVenus/+; R72F11-Gal4.AD/+; R38H09-Gal4.DBD/+* |
| 8 | D | SS04185 | *20xUAS-IVS-CsChrimson::mVenus/+; R72F11-Gal4.AD/R54B01-Gal4.AD; R38H09-Gal4.DBD/R46E07-Gal4.DBD* |
| 8 | E | ctrl | *20xUAS-IVS-CsChrimson::mVenus/+; R72F11-Gal4.AD/+; R38H09-Gal4.DBD/+* |
| 8 | E | SS04185 | *20xUAS-IVS-CsChrimson::mVenus/+; R72F11-Gal4.AD/R54B01-Gal4.AD; R38H09-Gal4.DBD/R46E07-Gal4.DBD* |
| 8 | F | control | *20xUAS-IVS-CsChrimson::mVenus/+; R72F11-Gal4.AD/+; R57F07-Gal4.DBD/+* |
| 8 | F | SS04185 | *20xUAS-IVS-CsChrimson::mVenus/+; R72F11-Gal4.AD/R54B01-Gal4.AD; R57F07-Gal4.DBD/R46E07-Gal4.DBD* |
| 8 | G | ctrl | *20xUAS-IVS-CsChrimson::mVenus/+; R72F11-Gal4.AD/+; R57F07-Gal4.DBD/+* |
| 8 | G | SS04185 | *20xUAS-IVS-CsChrimson::mVenus/+; R72F11-Gal4.AD/R54B01-Gal4.AD; R57F07-Gal4.DBD/R46E07-Gal4.DBD* |

## Behavior assay

To optogenetically stimulate neurons, embryos were collected for 24 hr and larvae were raised on fly food plates with 0.2 mM trans-retinal (Toronto Research Chemicals, R240000). The larvae were kept in the dark at 25°C for 4 days to grow to the third instar stage. Before the experiment, food plates with larvae were rinsed with a 15% sucrose solution to separate the larvae from the food. Larvae were then moved to a sieve, washed with water, dried, and placed evenly on 2% agar plates. The agar plate with animals were placed under a camera in the arena of the behavior rig.

### Behavior apparatus

The behavior rig consisted of several apparatuses (see *Ohyama et al., 2013* for details and modified by following), including a C-MOS camera (Grasshopper Camera USB3, GS3-U3-41C6M-C, FLIR), infrared 850 nm LED illumination (Waveform Lighting Co), a 624 nm (LED, Waveform Lighting Co), for optogenetic manipulations, a computer, and a heating panel. Both the camera and LED source were controlled by the computer. LED stimuli were controlled by customized software while larval behaviors were recorded using the Multi-Worm Tracker (MWT) software, a real-time image-analysis software (*Swierczek et al., 2011*). These two pieces of software were synchronized in the behavior assay to precisely deliver the stimulation during specified time windows.

## Optogenetic stimulation

Before delivering optogenetic stimulation, larvae were placed in the arena for 45 s. Subsequently, two 30 s 624 nm LED stimuli were presented successively with a 30-s interval between them. The LED intensity used in each experiment is shown below.

| Figure number | Optogenetic stimulation irradiance (µW/mm²) |
|---|---|
| *Figure 1* | 0.84 |
| *Figure 1—figure supplement 1* | 0.84 |
| *Figure 1—figure supplement 2* | 0.48 |
| *Figure 2C–E* | 5.9 |
| *Figure 2F–H* | 1.8 |
| *Figure 2—figure supplement 1B, C* | 5.9 |
| *Figure 2—figure supplement 1D, E* | 0.84 |
| *Figure 2—figure supplement 1H–J* | 1.8 |
| *Figure 5B–D* | 1.8 |
| *Figure 6* | 1.8 |
| *Figure 6—figure supplement 1* | 1.8 |
| *Figure 7A, B* | 0.84 |
| *Figure 7—figure supplement 1A–D* | 0.84 |
| *Figure 8A–D* | 3.9 |
| *Figure 8E–H* | 1.8 |
| *Figure 8—figure supplement 1A–C* | 1.8 |
| *Figure 8—figure supplement 1D, E* | 3.9 |
| *Figure 8—figure supplement 1F, G* | 1.8 |

## Thermal stimulation

To provide heat stimulation, we built thermal control systems with a proportional-integral-derivative temperature controller (ITC-106VH, Inkbird), a solid-state relay for temperature controllers (SSR-25A, Inkbird), a K-Type thermocouple to detect temperature, and a heat panel. The thermal control system was connected to a custom-built incubator designed to maintain a steady temperature inside the behavior rig at 32°C and warm the agar plates. The temperature of the agar plates was monitored by a thermometer gun (62 MAX+ Compact Infrared Thermometer, Fluke) before and after the experiment to verify the appropriate temperature for *shibire*^ts1 to be functional. Larvae were sealed in a plastic sieve and pre-heated in a water bath for 10 min to reach 32°C before the test. In order to maintain the temperature above 30°C during the test, a replica of the thermal control system mentioned above was installed in the behavior rig, and the behavior rig was pre-heated overnight before any thermal experiment.

For *shibire*^ts1 experiments with heat stimulation, during the first 5 s of the test, larvae were left on the agar plates without LED stimulation. Subsequently, the larvae were optogenetically stimulated with a 624-nm LED for 30 s.

## Behavior analysis

Larvae were tracked in real-time using Multi Worm Tracker (MWT) software (*Swierczek et al., 2011*, http://sourceforge.net/projects/mwt). The contour, spine, and center of mass for each larva were generated and recorded by MWT as a function of time. From these tracking data, the key parameters of larval motion were computed using Choreography software (a component of the MWT software package which measured the behavioral parameters offline) as described previously (*Ohyama et al., 2013*; *Ohyama et al., 2015*). The behavioral parameters generated by the Choreography algorithm

include speed, crabspeed (i.e., speed perpendicular to the body axis), body curvature, angle of head bending, body length, body width, area (dorsal view), and bias (i.e., fractional excess of time spent moving in one direction). In this offline process, objects that were tracked for less than 5 seconds or moved less than one body length of a larva were rejected. We refer the reader to the open-source package for further details of the software implementations for the above calculations.

## Behavior detection

After extracting behavioral parameters from Choreography, we used an unsupervised machine learning behavior classification algorithm to detect and quantify the following behaviors: hunching (Hunch), head-bending (Turn), stopping (Stop), and peristaltic crawling (Crawl) as previously reported (*Masson et al., 2020*). Escape rolling (Roll) was detected with a classifier developed using the Janelia Automatic Animal Behavior Annotator (JAABA) platform (*Kabra et al., 2013*; *Ohyama et al., 2015*). The rolling classifier is available at https://github.com/Jiayi2019/2024_elife (copy archieved at *Zhu, 2024a*). JAABA transforms the MWT tracking data into a collection of 'per-frame' behavioral parameters and regenerates two-dimensional dorsal-view videos of the tracked larvae. Based on such videos, we defined rolling as a rotation around the body while the larva maintains a C-shape, which results in a movement perpendicular to larval body axis (*Videos 1 and 2*). Using this definition, we trained the algorithm in the JAABA platform by labeling ~10,000 randomly chosen frames as rolling or non-rolling to develop the rolling classifier. If a larva did not curl into a C-shape or move sideways, it was labeled as a 'non-roller'. Every animal with at least one rolling event longer than 0.2 s in a given period was labeled as a 'roller' (i.e., it was assumed to have rolled at least 360 degrees), based on the observation that when the start and end of rolling events were precisely measured, the algorithm could identify rolling events completed in 0.2 s.

The rejection of false positives, especially at the beginning and the end of each rolling bout, enhanced accuracy. The algorithm integrated these training labels and parameters generated with Choreography in a time series, such as speed, crabspeed, and body curvature, to generate a score for rolling detection. Above a certain threshold, the classifier labeled the frame as rolling. This classifier, which has false-negative and -positive rates of 7.4% and 7.8%, respectively (*n* = 102), was utilized to detect rolling in this paper.

## Behavior quantification

The outputs of these behavior detection pipelines served as the input to a customized follow-up MATLAB-based analysis. Only the larvae being tracked fully during the stimulation window were selected for analysis. The percentages of animals performing given behaviors as well as their crawling speed in time series at a frame rate of 10 fps were plotted to depict the behavioral responses. To quantify the behavioral phenotype at the population level, the proportions of larvae that performed given behaviors at least once in the first 5 s after the onset of the stimulation were calculated in percentages. A collection of individual-level parameters (e.g., aggregated durations of rolling throughout the stimulation window, starts and ends of the first rolling event after stimulus onset, starts of the first crawling event after the first rolling event in the stimulation window) were generated and analyzed to describe the effects of stimulation on escape behaviors. Specifically, the starts of the first crawling events after the first rolling events were recorded as 30 s by default if larvae rolled but never initiated crawling during the stimulation window. Furthermore, the cumulative plots of the durations of each rolling event were contrasted to describe the event-level differences.

## **Larval dissections and immunohistochemistry**

Standard immunocytochemical procedures were followed (*Patel, 1994*). Briefly the CNSs of *Drosophila* larvae were dissected in phosphate-buffered saline (PBS). After dissection, tissues were fixed with 4% paraformaldehyde for 20 min, washed with PBS three times and then washed with 0.4% Triton X-100 in PBS (PBST) twice. Samples were incubated at room temperature with a blocking solution (5% normal goat serum [NGS]) for 1 hr. Next, the samples were incubated with the primary antibody solutions at 4°C overnight and washed for 15 min six times. Specially, anti-VGAT was incubated for 48 hr to compensate for the permeability. The primary antibodies were diluted at concentrations of 1:3000 for chicken anti-GFP; 1:1000 for rabbit anti-GFP, rabbit anti-GABA, and rabbit anti-dsRed; 1:200 for rabbit anti-VGAT; 1:50 for rat anti-Elav, 1:50 for mouse nc82; and 1:20 for rat anti-DN-Cadherin,

mouse anti-Fas2, mouse anti-choline acetyltransferase (ChAT), and rabbit anti-VGLUT in 5% NGS. CNS samples were then incubated with a secondary antibody solution at 4°C overnight and washed 15 min for six times. The secondary antibodies, including anti-chicken Alexa488, anti-rabbit Alexa488, anti-mouse Alexa568, anti-rabbit Alexa568, and anti-rat Alexa568, were all diluted at the concentration of 1:500. These samples were mounted in VECTASHIELD antifade mounting medium and imaged by a Zeiss LSM 710 confocal microscope with a 20×/NA0.8 objective lens (Zeiss) and Zen digital imaging software (Zeiss). For quantifying the expression of VGAT or GABA expression, laser power and gains are consistent between samples. All images were processed using Fiji software (https://imagej.new/ Fiji, ImageJ, NIH Bethesda).

### Immunohistochemistry image analysis

Larval CNS image stacks were processed with FIJI. For *Figure 5—figure supplement 1C,D*, four to six slices along the *z* dimension were averaged. The neuropil for VGAT or GABA channels at segments A4 to A6 and cell body regions for elav staining were manually selected as regions of interest (ROIs). The mean intensity of VGAT or GABA was measured and normalized by the mean value of elav staining.

### Two-photon calcium imaging assay

The CNSs of third instar larvae were dissected out in cold Baines external physiological saline (135 mM NaCl, 5 mM KCl, 5 mM TES, 36 mM sucrose, 2 mM $CaCl_2 \cdot 2H_2O$, 4 mM $MgCl_2 \cdot 6H_2O$, pH 7.15), and secured on a poly-L-lysine coated cover glass placed in a small Sylgard plate.

Functional calcium imaging experiments were performed on a customized two-photon microscope equipped with a Galvo-Resonant Scanner (Cambridge) controlled by Scanimage software (mbf BIOSCIENCE) using a 40×/0.80NA water immersion objective (LUMPlanFL, Olympus). A Mai Tai, Ti:Sapphire Ultrafast Laser (Spectra Physics) tuned to 925 nm was used for excitation of GCaMP protein. Fluorescence signals were collected with photomultiplier tubes (Hamamatsu) after bandpass filtering. Images were acquired by the Galvo-Resonant Scanner for a single plane of the CNS.

Each larva was stimulated by a 620-nm LED (Thorlabs) through the objective three times with a 30-s interval between periods of stimulation. Every stimulus consisted of a 30-ms pulse given every 100ms for a total of 1 s. Light intensity was measured to be 0.8–1.4 $\mu W/mm^2$. Images were acquired at a resolution of 512 × 512 pixels with a frame rate of 30 fps. Fluorescence intensities were averaged to 6 fps and processed in FIJI, and analyzed in MATLABwith customized scripts (available at https:// github.com/Jiayi2019/2-photon-analysis; copy archieved at *Zhu, 2024b*). ROIs were determined by the standard deviation of the full recording. $\Delta F = (F - F_0)/F_0$. $F_0$ is the average of images taken 10 frames (i.e., 1.7 s) before stimulation. $F$ is the mean value of the fluorescence in the ROI averaged every five frames from the start of the 5 s period before stimulation to end of the 15 s period after the onset of each stimulation. For each larva, $\Delta F$ is obtained through averaging the $\Delta F$ during the three stimulation periods. The peak $\Delta F$s were the maximal values selected from the onset of stimulation to 15 s after stimulus onset.

### Statistics

The probabilities for each response were analyzed by Chi-square tests. For the other parameters, when multiple groups were tested, their normality was examined first. If the normality assumption was rejected, Kruskal–Wallis tests were performed for multiple group variance comparisons, followed by multiple-comparison-corrected Wilcoxon–Mann–Whitney *U* tests as post hoc pairwise comparisons. If normality was met, analysis of variance was performed for variance comparisons and multiple-comparison-corrected Student's *t*-tests were utilized for pairwise comparisons. For two group comparisons, the Wilcoxon–Mann–Whitney *U* test was conducted if the normality assumption was offended, and the Student's *t*-test was applied if normality was met. All analyses were conducted with MATLAB.

### Inclusion and diversity

One or more of authors of this paper self-identifies as a member of the LGBTYQ+ community.

## Acknowledgements

We thank JW Truman, A Cardona, and M Zlatic for generating and sharing the split Gal4 lines. J Truman and M Zlatic for constructive inputs on the project. Confocal images were collected at the McGill University Advanced Bio Imaging Facility (ABIF), RRID:SCR_017697. We also thank the Bloomington stock center for providing fly stocks. We thank members of Ohyama lab for critical comments on the manuscript. This work was funded by McGill University, the Natural Sciences and Engineering Research Council (NSERC, RGPIN/04781-2017), the Canadian Institute of Health Research (CIHR, PTJ-376836), the Fonds de recherche du Quebéc – Nature et technologies (FRQNT, 2019-N-25523), and the Canada Foundation for Innovation (CFI, CFI365333). JCB was supported by Fonds de recherche du Quebéc-Santé (FRQS) graduate training award.

## Additional information

### Funding

| Funder | Grant reference number | Author |
|---|---|---|
| Natural Sciences and Engineering Research Council of Canada | RGPIN/04781-2017 | Tomoko Ohyama |
| Canadian Institutes of Health Research | 20161 OPJT-376836 | Tomoko Ohyama |
| Fonds de recherche du Québec – Nature et technologies | FRQ-NT NC-255237 | Tomoko Ohyama |
| Fonds de Recherche du Québec - Santé | 295825 | Tomoko Ohyama |
| Canada Foundation for Innovation | 36533 | Tomoko Ohyama |
| Fonds de Recherche du Québec - Santé | | Jean-Christophe Boivin |

The funders had no role in study design, data collection, and interpretation, or the decision to submit the work for publication.

### Author contributions

Jiayi Zhu, Conceptualization, Data curation, Formal analysis, Writing - original draft, Writing - review and editing; Jean-Christophe Boivin, Jing Ning, Yi Q Zhao, Data curation; Alastair Garner, Data curation, Methodology; Tomoko Ohyama, Conceptualization, Supervision, Funding acquisition, Methodology, Writing - original draft, Writing - review and editing

### Author ORCIDs

Tomoko Ohyama ⬚ https://orcid.org/0000-0003-1697-1138

Reviewer #1 (Public review): https://doi.org/10.7554/eLife.93978.3.sa1
Reviewer #2 (Public review): https://doi.org/10.7554/eLife.93978.3.sa2
Reviewer #3 (Public review): https://doi.org/10.7554/eLife.93978.3.sa3
Author response https://doi.org/10.7554/eLife.93978.3.sa4

## Additional files

### Supplementary files
• MDAR checklist

### Data availability

All data generated or analyzed during this study are included in the manuscript and supporting files; source data files have been provided for all figures. The source data for the Calcium imaging are

available at https://doi.org/10.6019/S-BIAD1314. The Code used for analysis is available at GitHub: https://github.com/Jiayi2019/ (copy archived at *Zhu, 2024a*; *Zhu, 2024b*).

The following dataset was generated:

| Author(s) | Year | Dataset title | Dataset URL | Database and Identifier |
|---|---|---|---|---|
| Ohyama T | 2024 | Calcium Imaging data for Zhu et. al 2024 elife | https://doi.org/10.6019/S-BIAD1314 | BioImage Archive, 10.6019/S-BIAD1314 |

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
