## [Editor Report · eLife assessment]

The aim of this **important** study is to functionally characterize neuronal circuits underlying the escape behavior in *Drosophila* larvae. Upon detection of a noxious stimulus, larvae follow a series of stereotyped movements that include bending of their body, rolling and crawling away. This paper combines quantitative behavioral analyses, cell-type specific manipulations, optogenetics, calcium imaging, immunostaining, and connectomic analysis to provide **convincing** evidence of an inhibitory descending pathway that controls the switch from rolling to fast crawling behaviors of the larval escape response.

---

## [Referee Report · Reviewer #1 (Public review)]

Summary:

Zhu et al. set out to better understand the neural mechanisms underlying *Drosophila* larval escape behavior. The escape behavior comprises several sequenced movements, including a lateral roll motion followed by fast crawling. The authors specifically were looking to identify neurons important for the roll-to-crawl transition.

Strengths:

This paper is clearly written, and the experiments are logical and complementary. They support the author's main claim that SeIN128 is a type of descending neuron that is both necessary and sufficient to modulate the termination of rolling. In general, the rigor is high.

Weaknesses:

-This manuscript is narrowly focused on *Drosophila* larval escape behavior. It would be more accessible to a broader audience if this work were put into a larger context of descending control.

---

## [Referee Report · Reviewer #2 (Public review)]

Summary:

The authors have addressed the majority of my comments, and I believe the revised manuscript has improved significantly.

The escape behavior of *Drosophila* larvae includes rolling followed by fast crawling, but the neural mechanism of this sequence was unclear. The authors determined the function of SeIN128, a group of descending neurons that terminate rolling and shorten crawling latency. SeIN128 receives inputs from Basin-2 and A00c neurons, which facilitate rolling, and makes reciprocal inhibitory synapses onto Basin-2 and A00c. SeIN128 shows a delayed activity peak upon Basins or A00c stimulation. Gad staining indicates that SeIN128 neurons are GABAergic, and blocking of SeIN128 function caused increased rolling probability and prolonged rolling. RNAi knockdown of GABA receptors in Basins suggests that several GABA receptors, especially GABA-A-R, mediate the SeIN128 to Basins inhibition. Among Basins subtypes, both Basin-2 and Basin-4 facilitate rolling but SeIN128 specifically terminates rolling elicited by Basin-2 activation. Overall, SeIN128 forms a feedback inhibition ensemble with Basin-2 and A00c that terminates rolling and shifts the animal to crawling.

Overall, this study discovered a neural mechanism that serves as a switch from rolling to fast crawling behaviors in *Drosophila* larvae. It addressed important open questions of how neural circuits determine the sequence of locomotor behaviors and how animals switch from one behavior to another. Its results support the conclusions and are backed up with proper control experiments.

Strengths:

- The question (i.e., the neural circuitry of action selection) addressed by this study is important.

- Larval and adult *Drosophila* is a powerful model system in neuroscience study, with rich genetic tools, diverse behaviors, and well-studied nervous systems. This study makes good use of them.

- The experiments, analyses, and results are rigorous and support the major claims. This study combined multiple innovative approaches, such as automated, machine-learning-based behavioral assays, EM reconstruction of larval CNS neurons, and genetic manipulation of specific neurons. A wide range of control experiments enhanced the credibility of the results.

- The graphical representations are clear and mindfully arranged.

Weaknesses:

I believe "Corkscrew-like rolling" is not an accurate term for larval rolling. The neuromuscular basis of rolling was recently studied by Cooney et. al., showing that rolling is the circumferential propagation of muscle activity where all segments contract similarly and synchronously. So using another term instead of "Corkscrew-like rolling" may help.

---

## [Referee Report · Reviewer #3 (Public review)]

Summary:

Combining the behavioral assays with optogenetics, imaging, and connectome approaches, this meticulous study characterizes the underlying neuronal mechanisms of escape behavior in *Drosophila* larvae. The authors identify the neurons and provide convincing evidence to support their function in the roll-to-crawl locomotor transition.

Strengths:

It is a very thorough characterization of locomotor sequences in terms of underlying neural circuits. The findings shed light on investigating the analogous behaviors in other systems.

Weaknesses:

None. The authors have revised the article to improve the presentation and clarity.

---

## [Author Response]

The following is the authors’ response to the original reviews.

**Reviewer #1 (Recommendations For The Authors):**
In my opinion, the three most important controls (hopefully easy):(1) Include no ATR controls for optogenetic activation experiments (not all, just one or two, e.g., Figure 4B, C, or D, for the highest activation condition). The concern is that it can be quite hard to use light to both monitor neural responses while also using light to activate the function of other neurons.

We thank the reviewer for the suggestions. We use a 2-photon 910-nm laser (which does not activate Chrimson) for imaging of GCaMP and a 624-nm LED (which does not activate GFP) for Chrimson activation. Calcium (GCaMP) signals are detected by PMT during Chrimson activation. With this setup, we are able to image GCaMP signals without crosstalk during activation of Chrimson.

We performed calcium imaging in animals that were not fed ATR and found that SS04185 showed no response to LED stimulation at the strongest intensity (µW/mm) (New Figure 4 – figure supplement 1B).

(2) Demonstrate that their RNAi constructs do indeed knock down the intended target gene. They showed nicely in Figure 5A that SeIN128 expresses GABA. Presumably, these neurons also express VGAT. Is it possible to check the expression of VGAT after RNAi knockdown? The concern is that using only a single RNAi introduces the possibility of off-target effects. Using multiple RNAi lines for VGAT or other parts of the pathway would also alleviate this (minor concern).

We thank the reviewer for raising this point. We agree that using only one RNAi line (HMS02355) for VGAT in Figure 5A is a weakness.

Accordingly, we have performed additional experiments to quantify the effect of RNAi knockdown of VGAT using HMS02335 in all neurons, followed by subsequent immunostaining against GABA or VGAT. We found that both VGAT and GABA were significantly reduced in the neuropil (Figure 5 – figure supplement 1C and D). These data strongly suggest that HMS02355 knocks down VGAT and reduces GABA at axon terminals. We note that HMS02355 has been used previously for knocking down GABA signaling in the following studies.

(1) Kallman BR, Kim H, Scott K (2015). Excitation and inhibition onto central courtship neurons biases *Drosophila* mate choice. eLife 4:e11188. https://doi.org/10.7554/eLife.11188

(2) Zhao W, Zhou P, Gong C et al. (2019). A disinhibitory mechanism biases *Drosophila* innate light preference. Nat Commun 10, 124. https://doi.org/10.1038/s41467-018-07929-w

(3) Yamagata N, Ezaki T, Takahashi T, Wu H, Tanimoto H (2021). Presynaptic inhibition of dopamine neurons controls optimistic bias. eLife 10:e64907. https://doi.org/10.7554/eLife.6490

(3) Include genetic controls for their driver line.In Figure 1, it would be nice to see one half or the other half of their split GAL4 line in their manipulations. The concern is that perhaps the phenotype is coming from something unexpected in the genetic background.

We thank the reviewer for the suggestion. We have added half of the GAL4 lines (AD or DBD) as controls (New Figure 1 – figure supplement 2). We found that SS04185 showed reduction of rolling, whereas AD only or DBD only (split control) did not (half of the split lines).

In the discussion:It seems that activation of SS014185 has additional effects beyond what the authors have quantified. Specifically, larvae do not appear to re-initiate rolling in the same manner as Basin activation alone. Also, there appears to be an off-response, turning.

We appreciate the reviewer’s comments. We have included a section in the discussion to consider the differences patterns of rolling observed during joint stimulation of Basins and SS04185 and during stimulation of Basins alone, as well as the increase in turning following the offset of joint stimulation of Basins and SS04185 compared with stimulation of Basins alone (lines 464 to 481). Although the reasons for these differences are beyond the scope of the paper, we have added Figure 2 – figure supplement 1K, which shows that co-activation of SS04185-MB and Basins is sufficient to evoke turning following the offset of stimulation, suggesting that the increased turning may be due to the activation of SS04185-MB neurons and independent of SS04185-DN neurons.

The labeling of the Figure panels could be improved. In many places, it is not clear that Basins are being stimulated in the background, whereas in nearby panels, it is clearly labeled. This is confusing for the reader.

We thank the reviewer for the constructive suggestions. We have modified all relevant figures to read “Basins>Chrimson” above the pink line indicating the period of optogenetic activation.

**Reviewer #2 (Recommendations For The Authors):**
Claims, rigorousness, repeatability, and accuracy of terms.(1) In line 254, the authors suggest that the slow response of SeIN128 neurons is due to the input they receive from SEZ, but in line 453, they suggest it is due to axo-axonal connections. However, their evidence does not support one factor over the other. Overall, only the axo-axonal connection was strongly suggested in the discussion. The authors could clarify that the delay of SeIN128 activity may also be caused by multisynaptic connections involving SEZ or other neurons in the last section of the Discussion.

Although SeIN128 primarily receives inputs from the SEZ, it also receives inputs within the VNC from Basin-2 (Figure 4 – figure supplement 2). Specifically, in the VNC, the axons of SeIN128 make inhibitory synaptic contacts onto the axon of Basin-2, which in turn makes reciprocal excitatory contacts onto the axon of SeIN128, thereby forming a feedback loop. However, by the time we wrote the original discussion, we had inadvertently focused on the potential of the negative feedback loop formed by these axo-axonal synapses in the VNC to mediate the slow response of SeIN128, overlooking the possibility that other as yet unidentified pathways could convey Basin or A00c activity indirectly to SeIN128 dendrites in the SEZ. Therefore, we have revised the original text, which read “These data suggest that the main synaptic inputs onto SeIN128 neurons in the SEZ mediate the slow responses upon activation of Basins or A00c neurons” to “These data suggest that the delay of SeIN128 activity may be caused by multi-synaptic connections involving the SEZ or a feedback loop involving axo-axonal connections between SeIN128 and Basin-2 or A00c” (revised, Lines 259 and 261). Accordingly, we have also adjusted the relevant discussion section to be consistent with this change (Lines 460 and 466).

(2) Please clarify the following: How does the algorithm define rolling and crawling? Healthy larvae complete 360{degree sign} rolls, in each roll they rotate from dorsal up to dorsal up. It is possible that a larva rolls for an incomplete cycle and straightens up. Does the algorithm simply label individual frames as “roll”, “non-roll”, or “unknown”, and defines rolling by the existence of “roll” frames? If so, then larvae that rolled for 90{degree sign} and straightened would be counted as “rolling” though they failed to complete a full rolling bout. Also, how were “hunch” “turn” and “back” identified? Lastly, is there any manual quality control involved? Address this and related issues in the methods:a) Expand the description of the classifier algorithm.b) How are rolling and non-rolling animals defined in the "rolling%" assay? Were all "rolling" animals able to do at least one 360{degree sign} roll?c) How are "rolling duration" and "end of 1st rolling" defined? Is the algorithm able to distinguish different rolling bouts? In these two assays, were the animals rolled for <1 second (in total or their "first roll") able to complete a 360{degree sign} roll?

The Multi-worm Tracker (MWT) records only the contours of animals (no real video image data). Thus, the data fed into the classifier algorithm only includes features based on contour time-series data. The algorism uses movement perpendicular to the body axis—the characteristic feature of larval rolling—to classify rollers and non-rollers. Although the algorithm cannot determine whether a rolling event involves a rotation of more than 360 degrees, we ensure that rolling events are at least 360 degrees by removing any events that are shorter than 0.2 s (the minimum time to complete a 360-degree roll).

We have accordingly revised the section of “Behavior detection” relating to the behavior classification algorithm in the methods section as follows (Lines 600 to 620).

“After extracting behavioral parameters from Choreography, we used an unsupervised machine learning behavior classification algorithm to detect and quantify the following behaviors: hunching (Hunch), headbending (Turn), stopping (Stop), and peristaltic crawling (Crawl) as previously reported (Masson et al., 2020). Escape rolling (Roll) was detected with a classifier developed using the Janelia Automatic Animal Behavior Annotator (JAABA) platform (Kabra et al., 2013; Ohyama et al., 2015). JAABA transforms the MWT tracking data into a collection of ‘per-frame’ behavioral parameters and regenerates 2D dorsal-view videos of the tracked larvae. Based on such videos, we defined rolling as a rotation around the body while the larva maintains a C-shape, which results in a movement perpendicular to larval body axis (Supplementary videos 1 and 2). Using this definition, we trained the algorithm in the JAABA platform by labeling ~10,000 randomly chosen frames as rolling or non-rolling to develop the rolling classifier. If a larva did not curl into a C-shape or move sideways, it was labeled as a “non-roller.” Every animal with at least one rolling event longer than 0.2 s in a given period was labeled as a “roller” (i.e., it was assumed to have rolled at least 360 degrees), based on the observation that when the start and end of rolling events were precisely measured, the algorithm could identify rolling events completed in 0.2 s.

The rejection of false positives, especially at the beginning and the end of each rolling bout, enhanced accuracy. The algorithm integrated these training labels and parameters generated with Choreography in a time series, such as speed, crabspeed, and body curvature, to generate a score for rolling detection. Above a certain threshold, the classifier labeled the frame as rolling. This classifier, which has false negative and false positive rates of 7.4% and 7.8%, respectively (n = 102), was utilized to detect rolling in this paper.”

Readability of text(1) I suggest giving the SS04185 line and SeIN128 neuron common names that are easier to remember and follow (after mentioning their full name once).

We acknowledge the reviewer’s concerns. However, because SS04185 was initially named using the Janelia split-line pipeline, and SeIN128 was named independently in a more recent study (Ohyama et al., 2015), we have retained these designations in the present manuscript.

Figures and figure legends(1) It would help if the authors could put visual representations of rolling and crawling, such as a cartoon larva performing the rolling-crawling switch, and still frames of rolling and crawling of real larvae, especially in Figure 1. Also, please consider including a video of rolling and crawling in real larvae (preferably comparing control and experimental groups).

We appreciate the reviewer’s suggestion. We have added a cartoon of the behavioral sequence in Figure 1A, as well as a Figure 1 supplement video based on MWT data, which shows rolling followed by crawling.

(2) To give the reader a take-home message, it would help if the authors could make a simplified version of Figure 4A and put it at the end of the paper.

We thank the reviewer for the suggestion. To assist the reader, we have added schematics depicting how the circuit may function in panel I of Figure 8.

(3) In Figure 1A, add the text "activation " after the neuron names.

We have added “Chrimson” following “Basins>” to the new Figure 1B (old Figure 1A) and other figures (Figure 1C and D, Figure 5A, Figure 6A, and figure supplements).

(4) Figure 1G: a data point is misaligned (at the top of the graph).

We have aligned the data point accordingly.

(5) Figure 1B can benefit from a better design. If possible, please separate the crawling speed into an independent graph (or at least use a different line shape to code for crawling speed and indicate it on the in-graph legend). Is the speed of Basin/SS04185 co-activation studied?

We appreciate the reviewer’s suggestion. We have separated the plots for rolling and crawling speed into different panels (Figure 1C and D). As shown in Figure 1D, the crawling speed observed during coactivation of Basins and SS04185 was similar to that during activation of Basins alone.

(6) Figure S1 uses a different color-coding scheme from Figure 1. I suggest making the color coding consistent between figures.

We are grateful for the reviewer’s suggestion. We have adjusted the color-coding scheme accordingly.

(7) Line 692 (Figure 2 legend), "Killer Zipper" is misspelled as "Kipper Zipper". Out of curiosity, is there a way to remove or reduce SS04185-DN expression in the same manner as SS04185-MB reduction?

We have corrected the text in the legend for Figure 2. As for the reviewer’s question, we did attempt to reduce or abolish SS04185-DN expression with tsh-LexA and LexAop-Kip+ but found no effect. Other identified LexA constructs with SeIN128 expression, however, all showed SS04185-MB expression. Consequently, we could not use these constructs because they inhibit both SeIN128 and SS04185-DN.

(8) The color coding of Figure 2 (especially in D) makes it hard to distinguish between the brown and red groups.

We thank the reviewer for the suggestion. Accordingly, we have changed the color for the brown group to orange.

(9) In line 926 (Figure S2 legends), the description of F and G seems inverted.

We appreciate the reviewer for pointing out the error. We have revised the text from “(F) has only SS04185-

MB expression, and (G) has both SS04185-DN and SS04185-MB expression” to “(F) has both SS04185DN and SS04185-MB expression, and (G) has only SS04185-MB expression.”

(10) Figure 7B: which line does the top group of asterisks belong to?

The top group of asterisks indicates that each experimental group differs significantly (p < 0.001) from the control group. We have revised the figure to clarify the comparisons indicated by the asterisks in Figure 7B, as well as the figure legend below (Line 890-894).

“(B) Cumulative plot of rolling duration. Statistics: Kruskal-Wallis test: H = 69.52, p < 0.001; Bonferronicorrected Mann-Whitney test, p < 0.001 between control and the GABA-B-R11, GABA-B-R12 and GABAB-R2 RNAi groups, p < 0.001 between GABA-A-R and all other experimental RNAi group. Sample size for the colored bars from top (control, black) to bottom (GABA-A-R, red); n = 520, 488, 387, 582, 306.”

(11) Figure S8 D and F: indicate Basin-2 or Basin-4 activation on graph.

We have revised Figure 8 – figure supplement D and F accordingly.

**Reviewer #3 (Recommendations For The Authors):**
(1) Lines 86-87: Text needs to be rewritten for clarity. Also, include the genotype in the corresponding figure legend (Figure 1B).

We thank the reviewer for pointing this out. We have clarified the text accordingly and included the genotype in the figure legend (lines 86 and 87). Specifically, we have revised Figure 1B (New Figure 1C and D) and adjusted the legend accordingly as follows.

Lines 86 and 87: Crawling speed during the activation of all Basins following rolling was ~1.5 times that of the crawling speed at baseline (Figure 1D).

(2) Include the protocol for heat shock-FLP out experiments

We have added the following paragraph to the Methods section describing the heat shock-FlpOut experiments (lines 537 to 546).

“Heat shock FlpOut mosaic expression

First instar *Drosophila* larvae were exposed to heat shock in a water bath at 37°C for 12 min as previously described (Nern et al., 2015). With precise temporal and thermal control of heat shock, larvae with genotype

w+, hs(KDRT.stop)FLP/13xLexAop2-IVS-CsChrimson::tdTomato; R54B01-Gal4.AD/72F11LexA;20xUAS-(FRT.stop)-CsChrimson::mVenus/R46E07-Gal4.DBD showed sporadic

CsChrimson::mVenus expression driven by SS04185 split GAL4. As a result, the ratio of the larvae with SS04185-DN and SS04185-MB expression to those with only SS04185-MB expression was 1:1. Each larva was individually examined with optogenetic stimulation and behavior analysis. After behavioral experiments, mVenus expression in CNS was confirmed under the fluorescence microscope.”

(3) In the immunohistochemistry, the authors exclude the steps for washings. Recommend the authors to cite the previous literature. Similar to the other protocols detailed in the methods.

We have added a brief description of the steps involved in washing (lines 641 and 648). We have also provided a citation with similar immunohistology protocols (Patel, 1994).

(4) Keeping the same Y-axis scale for similar graphical representation would be helpful to compare across different experimental conditions and genotypes-for example, 2E and 2H for the start of the first crawl.

As suggested by the reviewer, we have adjusted the y-axis scales for Figure 2E and H to be identical.

(5) The color schematics used for the graph make it hard to visualize the data. The author might reconsider the better presentation of the data by avoiding darker colors.

We thank the reviewer for the constructive suggestion. We have lightened the shading of all violin plots. We have also modified the shading for the middle group in Figure 2C and E from dark brown to orange.

(6) Co-activation of the SS04185 and Basins in the figures represented as Basins+SS04185 (Figure 1A) and SS04185 (rest of the figures). Authors might reconsider this terminology to define and distinguish the coactivation of SS04185 and Basins neurons from the activation of SS04185 or Basins alone. It needs to be clarified in the figures.

We have adjusted the terminology by including “Basins>Chrimson” in all panels in which Basin neurons are optogenetically activated to trigger rolling in the background for all groups. Additionally, we have labeled the control group as “Control” and the experimental group as ”SS04185”.

(7) Figure 4A, summarizes the synaptic connection and strength between different neurons - SeIN128, Basins, A00c and mdIV. However, the nature of these synaptic connections - excitatory and inhibitory- is not represented. Based on the previous and current studies, the authors consider providing the schematic for circuit mechanisms of escape behavior sequences in larvae. Also, discussing these findings in light of the downstream output circuit and motor regulation might be informative (See Cooney et al. 2023, PNAS).

As the reviewer correctly points out, the diagram of the connectome shown in Figure 4A does not indicate whether the connections are excitatory or inhibitory. Accordingly, we have added a new summary panel (Figure 8I) based on the results of examining GABAergic synapses (Figure 5A). The schematics in Figure 8I depict how the joint activity of inhibitory and excitatory synapses (indicated by arrowheads and blunt ends, respectively) may lead to rolling or fast crawling.

We have also added a section discussing the premotor circuits for crawling and rolling premotor circuit in discussion (Line 512 – 519).

(8) Percentage rolling present in figure 5B and 6A correspond to the control larvae 13xLexAop2-IVS-CsChrimson::mVenus; R72F11-lexA/+; HMS02355/+ and 13xLexAop2-IVS- Cs-Chrimson::mVenus; R72F11-lexA/+; UAS-TeTxLC.tnt/+. How does the author interpret the observed variability across the experiments? The author might consider discussing the genetic background effect on the observed behaviors, if any.

As pointed out by the reviewer, we noticed that rolling probability varied depending on genetic background. We have revised the text accordingly (Lines 277 to 280).

(9) Recheck the arrowheads in Figure 5A.

We have confirmed the positions of the arrowheads in Figure 5A and modified the figures by outlining the cells with dotted lines.

(10) Lines 295-298: Data presented in the supplementary figure and p-values in the text (p=0.11) suggest that the first crawl's onset is comparable to controls. Rewrite this text for clarity and include the statistical values in the supplemental figure 6.

We have revised the text as follows (Lines 302 to 305).

“Although the duration of each rolling bout, time to onset of the first rolling bout, and time to onset of the first crawling bout did not differ from those of controls (Figure 6–figure supplement 1D, E and G), the time to offset of the first rolling bout was delayed relative to controls (p = 0.013 for Figure 6–figure supplement 1F).”

(11) Lines 263-264: Data provide evidence for SS04185 receiving inputs Basin-2 and A00c neurons. SS04185, which provides inputs to other neurons, specifically A00c neurons, but still needs clarification.

We have revised the text as follows (Lines 264 to 266).

The results thus far indicate that, activation of SeIN128 neurons inhibits rolling (Figure 1A–C), SeIN128 neurons receive functional inputs from Basin-2 and A00c (Figure 4A-C); and SeIN128 neurons make anatomical connections onto Basin-2 and A00c (Figure 4A).

(12) In the table that lists the genotypes, instead of '-' or the blank space in the label column, the author might consider using 'control,' consistent with the figures.

In accord with the reviewer’s suggestion, we have revised the notation of ‘-’ or the blank space, to ‘control’ for all figures.

(13) Check the typographical errors throughout the manuscript. Some below:

We have revised the text accordingly as suggested below.

a. Lines 100, 142: SS4185 should be SS04185

b. Line 230: A00C should be A00c

c. Line 180: Expand VNC

d. 10xUAS-IVS-mry::GFP should be 10xUAS-IVS-myr::GFP

e. Lines 444, 449: *drosophila* should be *Drosophila*